# `AuroRA`: Breaking Low-Rank Bottleneck of LoRA with Nonlinear Mapping

**Haonan Dong[1], Wenhao Zhu[1], Guojie Song[†1], Liang Wang[2]**
[1]State Key Laboratory of General Artificial Intelligence,
School of Intelligence Science and Technology, Peking University,
[2]Alibaba Group, [†] Corresponding author
✉ hndong25@stu.pku.edu.cn, gjsong@pku.edu.cn

## Abstract

Low-Rank Adaptation (LoRA) is a widely adopted parameter-efficient fine-tuning (PEFT) method validated across NLP and CV domains. However, LoRA faces an inherent low-rank bottleneck: narrowing its performance gap with full fine-tuning requires increasing the rank of its parameter matrix, resulting in significant parameter overhead. Recent linear LoRA variants have attempted to enhance expressiveness by introducing additional linear mappings; however, their composition remains inherently linear and fails to fundamentally improve LoRA's representational capacity. To address this limitation, we propose `AuroRA`, which incorporates an Adaptive Nonlinear Layer (ANL) between two linear projectors to capture *fixed* and *learnable* nonlinearities. This combination forms an `MLP-like structure` with a compressed rank, enabling flexible and precise approximation of diverse target functions while theoretically guaranteeing lower approximation errors and bounded gradients. Extensive experiments on 22 datasets and 6 pretrained models demonstrate that `AuroRA`: **(I)** not only matches or surpasses full fine-tuning performance with only $6.18\% \sim 25\%$ of LoRA's parameters but also **(II)** outperforms competitive PEFT methods by up to $10.88\%$ in both NLP and CV tasks, and **(III)** exhibits robust performance across various rank configurations.

## 1 Introduction

In recent years, pretrained models have demonstrated excellent generalization performance across numerous tasks in various domains [1, 2, 3, 4, 5, 6]. In practical applications, to further unleash their powerful capabilities on specific downstream tasks, these models often require relevant fine-tuning [7, 8, 9, 10, 11, 12]. However, the increasing size of their parameters poses a significant challenge to fine-tuning all parameters [13, 14]. To address this issue, the field of Parameter-Efficient Fine-Tuning (PEFT) has made substantial progress [13, 15, 16, 17, 18, 19, 20]. The core idea is to fine-tune only a small subset of the model's parameters while freezing the majority of the pretrained parameters, achieving performance comparable to full fine-tuning [21].

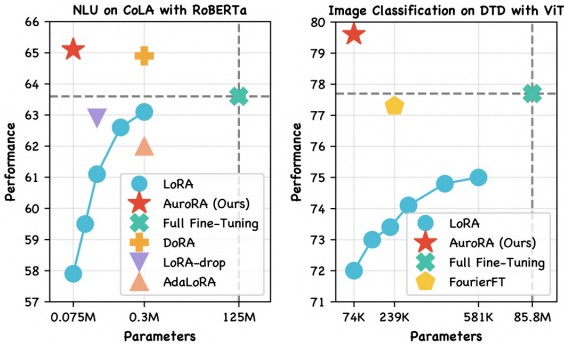

Figure 1: The trade-off between parameters and performance of various fine-tuning methods on NLP (left) and CV (right) tasks. (*Left*) In NLU, RoBERTa-Base is fine-tuned on COLA, with LoRA ranks $r = \{2, 3, 4, 6, 8\}$. (*Right*) In image classification, ViT-Base is fine-tuned on DTD, with LoRA ranks $r = \{2, 4, 6, 8, 12, 16\}$.

39th Conference on Neural Information Processing Systems (NeurIPS 2025).

LoRA is a commonly used state-of-the-art PEFT method [16]. Specifically, it assumes that the weight updates conform to a low-rank hypothesis and represents these updates using two low-rank matrices, i.e., $W_0 + \Delta W = W_0 + BA$. Its performance has been validated in fields such as natural language processing (NLP) [16, 22] and computer vision (CV) [23, 24]. Despite its significant success, LoRA still faces an inherent limitation, namely *the low-rank bottleneck*, as illustrated in Figure 1. As the rank of LoRA increases, the model's performance improves, thereby narrowing the gap with full fine-tuning [25]; however, the parameter cost grows proportionally with the rank, which weakens its parameter efficiency. This dilemma leads us to the first research question: ❶ *Can we achieve a further balance between parameters and performance?*

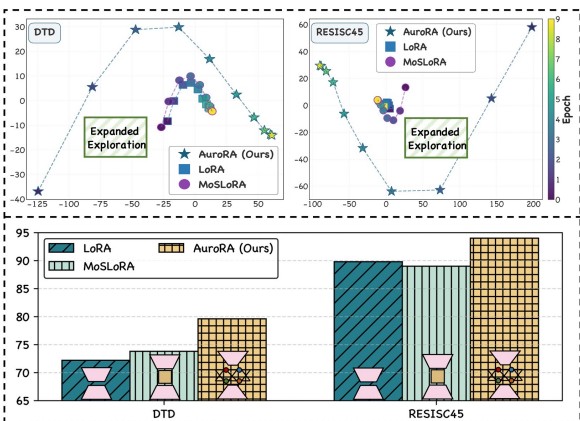

Figure 2: We evaluate LoRA, MoSLoRA, and our `AuroRA` on DTD and RESISC45 datasets, employing ViT-Base with a rank of $r = 2$. (***Upper***) We record the $\Delta W$ at the $\{0, 1, 2, ..., 9\}$-th epochs, and perform PCA visualization on these $\Delta W$. We observe that `AuroRA` is capable of exploring a broader parameter space. (***Lower***) We present the accuracy results on both datasets.

Recently, several linear LoRA variants have emerged [26, 27, 28, 29]. They introduce an *additional matrix* between the $B$ and $A$ matrices of LoRA to weaken the correlation constraints between them, thereby enhancing LoRA's learning and expressive capabilities. Specifically, one approach involves the introduction of a *diagonal matrix* to facilitate singular value decomposition [26, 27], while another approach incorporates an *arbitrary matrix* to fuse subspaces [28, 29]. Nevertheless, LoRA's inherent linearity persists even when an additional matrix is introduced, preserving its fundamental structure as a linear mapping. As illustrated in Figure 2, when the rank is extremely low, MoSLoRA [28] (a linear variant that incorporates an arbitrary matrix) has only a *marginal* effect on expanding the exploration of $\Delta W$, leading to a failure to further boost performance ($2.2\% \uparrow$ on DTD and $0.56\% \downarrow$ on RESISC45). The structural characteristics and resultant performance limitations of linear variants naturally prompt our second research question: ❷ *Can we achieve more than marginal performance improvements by introducing a nonlinear transformation between LoRA's two linear layers?*

Motivated by the above two research questions, this paper focuses on introducing nonlinear mappings into LoRA and further compressing the rank to achieve a better balance between parameters and performance. To this end, we propose a method called Activate Your Low-Rank Adaptation (`AuroRA`). We revisit LoRA through the lens of linear mappings and identify two critical limitations: (**I**) *insufficient expressiveness* and (**II**) *limited training flexibility*. To fully harness LoRA's potential, `AuroRA` introduces an **A**daptive **N**onlinear **L**ayer (**ANL**) between the low-rank matrices, forming an **MLP-like structure**. ANL employs a hybrid design of *fixed* and *learnable* nonlinearities to enhance model expressivity within a more compressed rank while enabling flexible training strategies to expand the explorable parameter space (Figures 1 and 2). Theoretical analysis demonstrates that `AuroRA` not only achieves a strictly lower approximation error than LoRA but also preserves bounded gradient norms. Experiments across NLP and CV tasks confirm the *efficiency*, *generalizability*, and *robustness* of `AuroRA`. We further conduct ablation studies to dissect the contributions of fixed and learnable components, and evaluate its robustness against linear LoRA variants across multiple rank configurations. Our contributions can be summarized as follows:

❶ *Perspective Shift.* We systematically revisit two research lines of LoRA: *the low-rank bottleneck* and *linear LoRA variants*. By interpreting LoRA through the lens of linear mappings, we address both research questions within a unified framework, providing theoretical analyses.

❷ *Nonlinear Proposal.* We propose `AuroRA`, which introduces nonlinear mappings into LoRA and further compresses the rank, resulting in a superior balance between parameters and performance, paving the way for further unlocking the significant potential of LoRA.

❸ *Experimental Validation.* Extensive experiments on 22 datasets and 6 pretrained models showcase that `AuroRA`: (**I**) not only matches or surpasses full fine-tuning performance with only $6.18\% \sim 25\%$ of LoRA's parameters but also (**II**) outperforms competitive PEFT methods by up to $10.88\%$ in NLP and CV tasks, and (**III**) exhibits robust performance across various rank configurations.

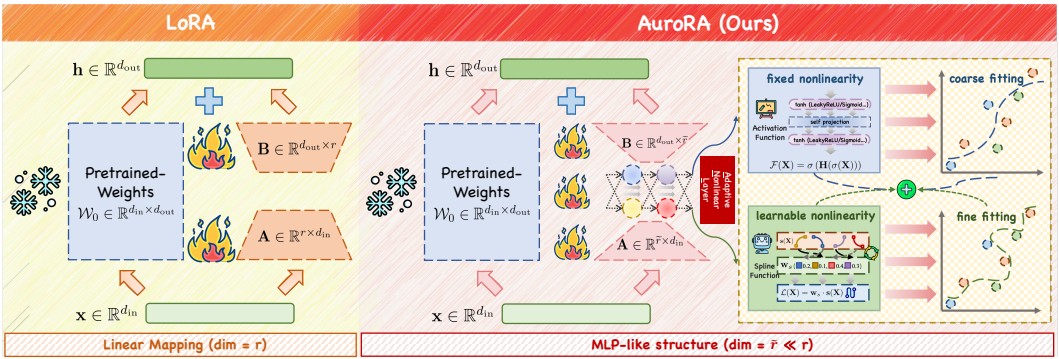

Figure 3: A general comparison of LoRA and our `AuroRA`. (*Left*) In LoRA, matrices $\mathbf{A}$ and $\mathbf{B}$ act as two linear projectors, forming a two-layer linear mapping with hidden dimension $r$. (***Right***) Our `AuroRA` extends LoRA by incorporating an adaptive nonlinear layer comprising fixed and learnable nonlinearities, forming an MLP-like structure with significantly reduced hidden dimension $\widetilde{r}$ ($\widetilde{r} \ll r$).

## 2   Methodology

As illustrated in Figure 3, we introduce `AuroRA`, an extension of LoRA that incorporates nonlinear mappings to overcome the inherent low-rank bottleneck. We reinterpret LoRA as a **two-layer linear mapping**, whereas our proposed `AuroRA` transforms it into an **MLP-like structure** by introducing an adaptive nonlinear layer.

### 2.1   LoRA: A Two-Layer Linear Mapping

In standard LoRA [16], the weight update $\Delta\mathcal{W}$ for a pre-trained weight matrix $\mathcal{W}_0$ is approximated as the product of two low-rank matrices:

$$\Delta\mathcal{W} = \mathbf{BA}, \tag{1}$$

where $\mathbf{A} \in \mathbb{R}^{r \times d_{\text{in}}}$ and $\mathbf{B} \in \mathbb{R}^{d_{\text{out}} \times r}$, with the rank $r$ satisfying $r \ll \min(d_{\text{in}}, d_{\text{out}})$. The forward propagation for an input vector $\mathbf{x} \in \mathbb{R}^{d_{\text{in}}}$ is thus expressed as:

$$\mathbf{h} = \mathcal{W}_0\mathbf{x} + \Delta\mathcal{W}\mathbf{x} = \mathcal{W}_0\mathbf{x} + \mathbf{BAx}. \tag{2}$$

The above process can be interpreted as a two-layer linear mapping, where $\mathbf{A}$ serves as a downward projector $\mathcal{P}_{\text{down}}$ that maps the input $\mathbf{x}$ from a high-dimensional space $\mathbb{R}^{d_{\text{in}}}$ to a lower-dimensional hidden space $\mathbb{R}^r$, and $\mathbf{B}$ serves as an upward projector $\mathcal{P}_{\text{up}}$ that maps back to $\mathbb{R}^{d_{\text{out}}}$. However, we note that LoRA is constrained by its sequential linear mapping structure, leading to two significant shortcomings: ❶ **insufficient expressiveness**: being a purely linear structure, it requires increasing the hidden dimension to handle more complex incremental weights and improve performance; ❷ **limited training flexibility**: the direct low-rank decomposition induces strong interdependencies between the linear layers, imposing rigid structural constraints that reduce training flexibility [30].

### 2.2   `AuroRA`: An MLP-like Structure

To address these limitations, `AuroRA` introduces an Adaptive Nonlinear Layer (ANL) between $\mathbf{A}$ and $\mathbf{B}$, modifying the weight update as follows:

$$\Delta\mathcal{W} = \mathbf{B} \cdot \sigma(\mathbf{A}), \tag{3}$$

where $\sigma$ is the element-wise ANL that maps from $\mathbb{R}^{\widetilde{r}}$ to $\mathbb{R}^{\widetilde{r}}$. Here, $\widetilde{r}$ denotes the compressed hidden dimension ($\widetilde{r} \ll r$), i.e., the low dimension to which the input is projected by $\mathcal{P}_{\text{down}}$. Formally, the forward propagation equation in the training phase is given by:

$$\mathbf{h} = \mathcal{W}_0\mathbf{x} + \mathbf{B} \cdot \sigma(\mathbf{Ax}). \tag{4}$$

The introduction of ANL enables `AuroRA` to form an MLP (Multilayer Perceptron)-like structure.

### 2.3   Adaptive Nonlinear Layer

Consider an arbitrary input vector $\mathbf{z}$. After projecting $\mathbf{z}$ into an $\widetilde{r}$-dimensional hidden layer, our objective is to introduce sufficient nonlinearity to capture as many complex relationships as possible

within this limited hidden space. To achieve this, we propose the following components: ❶ *fixed nonlinearity ($\mathcal{F}$)*, which utilizes parameter-free nonlinear activation functions to activate neurons in the hidden space, thereby achieving *coarse fitting*; and ❷ *learnable nonlinearity ($\mathcal{L}$)*, which employs parameterized nonlinear functions during the training process of the weight update increments, facilitating *fine fitting*. By combining ❶ and ❷, the Adaptive Nonlinear Layer (ANL) can be formally expressed as:

$$\sigma(\mathbf{Z}) = \mathcal{F}(\mathbf{Z}) + \mathcal{L}(\mathbf{Z}), \tag{5}$$

where $\mathcal{F}$ represents the fixed nonlinear activation, $\mathcal{L}$ denotes the learnable nonlinear function, and $\mathbf{Z}$ denotes the input to ANL. We provide a detailed comparison of fixed and learnable nonlinearity in Section 3.4.

For ❶, we adopt widely used activation functions in deep learning, such as ReLU [31], sigmoid, and tanh. A detailed comparison of different activation functions and their impact on `AuroRA`'s performance is provided in Section 3.4. Through our comparative evaluations, tanh emerges as the top-performing activation function, and theoretical analysis concurrently ensures its training stability. This preference is consistent with empirical findings in prior studies [32, 33] that demonstrate the robust performance of the tanh activation function for large-scale models, leading us to employ tanh in our implementation. The depth of the network influences the number of activation functions that can be introduced. Specifically, we introduce a *self-projection* $\mathcal{P}_{\text{self}} \in \mathbb{R}^{\widetilde{r} \times \widetilde{r}}$ between $\mathcal{P}_{\text{down}}$ and $\mathcal{P}_{\text{up}}$, which extends the depth of the standard LoRA structure. Subsequently, we introduce tanh activation functions between $\mathcal{P}_{\text{down}}$ and $\mathcal{P}_{\text{self}}$, and between $\mathcal{P}_{\text{self}}$ and $\mathcal{P}_{\text{up}}$. Formally, the fixed nonlinear component is defined as:

$$\mathcal{F}(\mathbf{Z}) = \tanh\left(\mathbf{H}\left(\tanh(\mathbf{Z})\right)\right), \tag{6}$$

where $\mathbf{H} \in \mathbb{R}^{\widetilde{r} \times \widetilde{r}}$ denotes $\mathcal{P}_{\text{self}}$.

To achieve ❷, we propose using spline functions to model complex relationships [34]. Numerous prior studies [35, 36, 37, 38] have demonstrated that splines are flexible, piecewise polynomial functions capable of approximating a wide range of nonlinear behaviors. Specifically, we employ B-spline basis functions to construct the learnable component. Formally, the learnable nonlinear component is defined as:

$$\mathcal{L}(\mathbf{Z}) = \mathbf{w}_s \cdot \mathbf{s}(\mathbf{Z}), \tag{7}$$

where $\mathbf{w}_s \in \mathbb{R}^{\widetilde{r}}$ is the spline weight vector, and $\mathbf{s}(\mathbf{Z}) = \sum_{i=1}^{\widetilde{r}} B(z_i)$ represents the spline basis functions applied to each dimension $z_i$ of $\mathbf{Z}$. The learnable parameters in this component are the spline weights $\mathbf{w}_s$, which determine the contribution of each basis function $B(z_i)$ to the overall output of $\mathcal{L}(\mathbf{Z})$. During training, these weights are iteratively updated to minimize the task-specific loss function.

By introducing ❶ and ❷ in the hidden layer with dimension $\widetilde{r}$, ANL effectively captures complex relationships without significantly increasing the number of additional parameters. The combination of *coarse fitting* and *fine fitting* enhances the standard LoRA structure, improving its expressive capacity and training flexibility, achieving what we refer to as `Activate Your Low-Rank Adaptation`. The complete Adaptive Nonlinear Layer (ANL) developed in our work can then be formally represented as:

$$\sigma(\mathbf{Z}) = \tanh\left(\mathbf{H}\left(\tanh(\mathbf{Z})\right)\right) + \mathbf{w}_s \cdot \mathbf{s}(\mathbf{Z}). \tag{8}$$

Further details and the complete algorithmic workflow of `AuroRA` are provided in Appendix B.

## 2.4 Theoretical Analysis

In this subsection, we propose two theoretical propositions concerning `AuroRA` and analyze its parameter and computational cost. Additionally, we present an intuitive case in Appendix C to help better understand the role of nonlinearities.

**Proposition 2.1** (Lower Approximation Error). *Let $M \in \mathbb{R}^{d_{\text{out}} \times d_{\text{in}}}$ with $\text{rank}(M) > r$. Define*

$$\varepsilon_r(M) = \inf_{U \in \mathbb{R}^{d_{\text{out}} \times r}, V \in \mathbb{R}^{r \times d_{\text{in}}}} \| M - UV \|.$$

*Then $\varepsilon_r(M) > 0$, and for our proposed update of the form*

$$M_{\text{nonlinear}}(\mathbf{x}) = B \, \sigma\left(A \, \mathbf{x}\right), \quad A \in \mathbb{R}^{r \times d_{\text{in}}}, \ B \in \mathbb{R}^{d_{\text{out}} \times r},$$

*where $\sigma$ is our adaptive nonlinear layer, there exists a parameter set $(A^*, B^*, \sigma^*)$ such that*

$$\left\| M - M_{\text{nonlinear}} \right\| \le c \, \varepsilon_r(M), \quad 0 < c < 1.$$

*Hence, the approximation error is strictly below the linear rank-r limit $\varepsilon_r(M)$, using the same rank r.*

▶ Proposition 2.1 indicates that, thanks to the introduction of nonlinear mappings, `AuroRA` achieves a strictly lower approximation error compared to LoRA at the same rank, meaning that the resulting weight updates are closer to the optimal solution. Furthermore, our empirical results demonstrate that this improvement persists even when further compressing the hidden dimensions of `AuroRA`. A rigorous proof, along with technical details and error bounds, is provided in Appendix D.

**Proposition 2.2** (Gradient Boundedness). *In the `AuroRA`, the use of the $\tanh$ activation function and B-spline basis functions results in bounded gradients with respect to both the inputs and the model parameters.*

▶ Proposition 2.2 posits that, despite the introduction of fixed and learnable nonlinearities, `AuroRA` maintains bounded gradients during training, thereby ensuring training stability. The corresponding proof is provided in Appendix E.

**Parameter Cost**  In Section 1, we discussed the relationship between trainable parameters and rank in LoRA, where the number of introduced trainable parameters is $O(r(d_{\text{in}} + d_{\text{out}}))$. Here, $d_{\text{in}}$ and $d_{\text{out}}$ represent the input and output dimensions, respectively, i.e., PARAMS $\propto r$. In `AuroRA`, we aim to further compress the parameter count by setting the hidden layer dimension to $\widetilde{r} = r/k$, where $k$ is a constant and $r$ is the optimal rank setting of LoRA. This means that the number of trainable parameters in `AuroRA` is $1/k$ of that in LoRA. In this work, we set $\widetilde{r}$ to 2, corresponding to values of $k$ such as 4 and 8. The additional parameters introduced in ANL are of the order $O(2\widetilde{r}^2)$, which, compared to the significant reduction in parameter count, can be considered negligible.

**Computational Cost**  The computational complexity of `AuroRA`'s forward pass in the training phase, $\Delta\mathbf{h} = \mathbf{B}\sigma \cdot (\mathbf{A}\mathbf{x})$, is analyzed as follows. Let $b$ denote the batch size, $d_{in}$ and $d_{out}$ the input/output feature dimensions, $r$ the rank, and $G$ the collective B-spline parameters (a small constant, $G = O(r)$). The linear projections by $\mathbf{A} \in \mathbb{R}^{r \times d_{in}}$ and $\mathbf{B} \in \mathbb{R}^{d_{out} \times r}$ incur complexities of $O(bd_{in}r)$ and $O(brd_{out})$, respectively. The intermediate fixed and learnable non-linearities, $\sigma(\cdot)$, each contribute an additional $O(br^2)$ term (with the learnable component's $O(brG)$ complexity simplifying due to $G = O(r)$). Consequently, the total complexity for `AuroRA` is $O(b(d_{in}r + 2r^2 + rd_{out}))$. Given the standard low-rank setting where $r \ll \min(d_{in}, d_{out})$, the quadratic overhead $O(br^2)$ introduced by the non-linearities is negligible compared to the dominant linear terms, thus maintaining a computational footprint comparable to that of LoRA.

## 3 Experiments

In this section, we conduct extensive experiments to answer the following research questions: ($\mathcal{RQ}$1) Can `AuroRA` effectively achieve efficiency in NLP tasks? ($\mathcal{RQ}$2) Can `AuroRA` effectively achieve efficiency in CV tasks? ($\mathcal{RQ}$3) What are the respective roles of fixed and learnable nonlinearity? ($\mathcal{RQ}$4) How do different activation functions in fixed nonlinearity affect performance? ($\mathcal{RQ}$5) How does `AuroRA`'s sensitivity to rank compare to that of linear LoRA variants? [1]

### 3.1 Experimental Setup

#### 3.1.1 Datasets and Pre-Trained Models

**Datasets**  For our experiments, we evaluate the ability of `AuroRA` to achieve parameter-efficient fine-tuning using four categories of datasets spanning both NLP and CV domains: ■ **Natural Language Understanding**: We employ GLUE (General Language Understanding Evaluation) [39], a widely used multi-task benchmark in NLU, which includes datasets such as SST-2, MRPC, CoLA, QNLI, RTE, and STS-B. The evaluation metrics are as follows: CoLA is assessed using Matthew's correlation coefficient, STS-B with Pearson's correlation coefficient, and accuracy is used for the other tasks. ■ **Commonsense Reasoning**: We use a collection of commonly used datasets, including BoolQ [40], PIQA [41], SocialIQA [42], HellaSwag [43], WinoGrande [44], ARC-e, ARC-c [45], and OpenBookQA [46]. For fair comparison, we follow the setup proposed by [28], fine-tuning the pretrained models on the Commonsense170K dataset, which serves as a mixture of the aforementioned benchmark datasets. We then evaluate using accuracy as the performance metric. ■ **Image Classification**: We use five datasets with small label spaces—OxfordPets [47], CIFAR-10 [48], DTD [49], EuroSAT [50], and RESISC45 [51], and three datasets with large label

---

[1]The source code is available at here.

Table 1: We report the performance of different fine-tuning methods on six datasets of the GLUE benchmark, using RoBERTa-Base and RoBERTa-Large models. For CoLA, we report the Matthew's Correlation Coefficient (MCC); for STS-B, we report the Pearson Correlation Coefficient (PCC); and for all other tasks, we report accuracy (Acc.). The reported results are the medians of five runs, each using a different random seed. * indicates numbers published in prior works. The best results are highlighted in **bold**, and the runners-up are underlined. For all six datasets, higher values are considered better for all metrics.

| Model | Method | SST-2 | MRPC | CoLA | QNLI | RTE | STS-B | Avg. | Params. |
|---|---|---|---|---|---|---|---|---|---|
| RoBERTa-Base | Full Fine-Tuning* | 94.8 | 90.2 | 63.6 | 92.8 | 78.7 | 91.2 | 85.2 | 125M |
| | BitFit* | $93.7_{\downarrow1.1}$ | $\mathbf{92.7_{\uparrow2.5}}$ | $62.0_{\downarrow1.6}$ | $91.8_{\downarrow1.0}$ | $\underline{81.5}_{\uparrow2.8}$ | $90.8_{\downarrow0.4}$ | $85.4_{\uparrow0.2}$ | 0.1M |
| | Adapter$^D$* | $94.7_{\downarrow0.1}$ | $88.4_{\downarrow1.8}$ | $62.6_{\downarrow1.0}$ | $93.0_{\uparrow0.2}$ | $75.9_{\downarrow2.8}$ | $90.3_{\downarrow0.9}$ | $84.2_{\downarrow1.0}$ | 0.9M |
| | LoRA* | $\underline{95.1}_{\uparrow0.3}$ | $89.7_{\downarrow0.5}$ | $63.4_{\downarrow0.2}$ | $\underline{93.3}_{\uparrow0.5}$ | $78.4_{\downarrow0.3}$ | $91.5_{\uparrow0.3}$ | $85.2_{\uparrow0.0}$ | 0.3M |
| | AdaLoRA* | $94.5_{\downarrow0.3}$ | $88.7_{\downarrow1.5}$ | $62.0_{\downarrow1.6}$ | $93.1_{\uparrow0.3}$ | $81.0_{\uparrow2.3}$ | $90.5_{\downarrow0.7}$ | $85.0_{\downarrow0.2}$ | 0.3M |
| | DyLoRA* | $94.3_{\downarrow0.5}$ | $89.5_{\downarrow0.7}$ | $61.1_{\downarrow2.5}$ | $92.2_{\downarrow0.6}$ | $78.7_{\uparrow0.0}$ | $91.1_{\downarrow0.1}$ | $84.5_{\downarrow1.3}$ | 0.3M |
| | FourierFT* | $94.2_{\downarrow0.6}$ | $90.0_{\downarrow0.2}$ | $63.8_{\uparrow0.2}$ | $92.2_{\downarrow0.6}$ | $79.1_{\uparrow0.4}$ | $90.8_{\downarrow0.4}$ | $85.0_{\downarrow0.2}$ | 0.024M |
| | LoRA-drop* | $94.5_{\downarrow0.3}$ | $89.5_{\downarrow0.7}$ | $62.9_{\downarrow0.7}$ | $93.1_{\uparrow0.3}$ | $81.4_{\uparrow2.7}$ | $91.0_{\downarrow0.2}$ | $85.4_{\uparrow0.2}$ | 0.15M |
| | DoRA* | $95.0_{\uparrow0.2}$ | $89.7_{\downarrow0.5}$ | $\underline{64.9}_{\uparrow1.3}$ | $92.9_{\uparrow0.1}$ | $79.2_{\downarrow0.5}$ | $\underline{91.3}_{\uparrow0.1}$ | $\underline{85.5}_{\uparrow0.3}$ | 0.3M |
| | AuroRA | $\mathbf{95.2_{\uparrow0.4}}$ | $\underline{91.9}_{\uparrow1.7}$ | $\mathbf{65.1_{\uparrow1.5}}$ | $\mathbf{93.4_{\uparrow0.6}}$ | $\mathbf{85.2_{\uparrow6.5}}$ | $\mathbf{91.5_{\uparrow0.3}}$ | $\mathbf{87.1_{\uparrow1.7}}$ | 0.075M |
| RoBERTa-Large | Full Fine-Tuning* | $\underline{96.4}$ | $\underline{90.9}$ | 68.0 | 94.7 | 86.6 | $\underline{92.4}$ | $\underline{88.2}$ | 356M |
| | Adapter$^P$* | $96.1_{\downarrow0.3}$ | $90.2_{\downarrow0.7}$ | $\underline{68.3}_{\uparrow0.3}$ | $\underline{94.8}_{\uparrow0.1}$ | $83.8_{\downarrow2.8}$ | $92.1_{\downarrow0.3}$ | $87.6_{\downarrow0.6}$ | 3M |
| | Adapter$^H$* | $96.2_{\downarrow0.2}$ | $88.7_{\downarrow2.2}$ | $66.5_{\downarrow1.5}$ | $94.7_{\uparrow0.0}$ | $83.4_{\downarrow3.2}$ | $91.0_{\downarrow1.2}$ | $86.8_{\downarrow1.4}$ | 6M |
| | LoRA* | $96.2_{\downarrow0.2}$ | $90.2_{\downarrow0.7}$ | $68.2_{\uparrow0.2}$ | $\underline{94.8}_{\uparrow0.1}$ | $85.2_{\downarrow1.4}$ | $92.3_{\downarrow0.1}$ | $87.8_{\downarrow0.4}$ | 0.8M |
| | FourierFT* | $96.0_{\downarrow0.4}$ | $\underline{90.9}_{\uparrow0.0}$ | $67.1_{\downarrow0.9}$ | $94.4_{\downarrow0.3}$ | $\underline{87.4}_{\uparrow0.8}$ | $91.9_{\downarrow0.5}$ | $88.0_{\downarrow0.2}$ | 0.048M |
| | AuroRA | $\mathbf{96.6_{\uparrow0.2}}$ | $\mathbf{91.2_{\uparrow0.3}}$ | $\mathbf{69.2_{\uparrow1.2}}$ | $\mathbf{95.0_{\uparrow0.3}}$ | $\mathbf{89.9_{\uparrow3.3}}$ | $\mathbf{92.5_{\uparrow0.1}}$ | $\mathbf{89.1_{\uparrow0.9}}$ | 0.2M |

Table 2: Commonsense reasoning evaluation results for LLaMA3-8B on eight tasks. * indicates numbers taken from [57]. The best results are highlighted in **bold**, and the runners-up are underlined. For all eight tasks, higher values are considered better.

| Method | Params. | BoolQ | PIQA | SIQA | HellaSwag | WinoGrande | ARC-e | ARC-c | OBQA | Avg. |
|---|---|---|---|---|---|---|---|---|---|---|
| LoRA* | 56.6M | $\underline{70.8}$ | 85.2 | $\mathbf{79.9}$ | 91.7 | $\underline{84.3}$ | 84.2 | 71.2 | 79.0 | 80.8 |
| PiSSA* | 83.8M | $67.1_{\downarrow3.7}$ | $81.1_{\downarrow4.1}$ | $77.2_{\downarrow2.7}$ | $83.6_{\downarrow8.1}$ | $78.9_{\downarrow5.4}$ | $77.7_{\downarrow6.5}$ | $63.2_{\downarrow8.0}$ | $74.6_{\downarrow5.4}$ | $75.4_{\downarrow5.4}$ |
| MiLoRA* | 56.6M | $68.8_{\downarrow2.0}$ | $\underline{86.7}_{\uparrow1.5}$ | $77.2_{\downarrow2.7}$ | $\underline{92.9}_{\uparrow1.2}$ | $\mathbf{85.6_{\uparrow1.3}}$ | $\underline{86.8}_{\uparrow2.6}$ | $\underline{75.5}_{\uparrow4.3}$ | $\underline{81.8}_{\uparrow2.8}$ | $\underline{81.9}_{\uparrow1.1}$ |
| AuroRA | 3.5M | $\mathbf{72.5_{\uparrow1.7}}$ | $\mathbf{87.4_{\uparrow2.2}}$ | $\underline{79.0}_{\downarrow0.9}$ | $\mathbf{94.2_{\uparrow2.5}}$ | $83.0_{\downarrow1.3}$ | $\mathbf{89.3_{\uparrow5.1}}$ | $\mathbf{78.8_{\uparrow7.6}}$ | $\mathbf{84.8_{\uparrow5.8}}$ | $\mathbf{83.6_{\uparrow2.8}}$ |

spaces, namely StanfordCars [52], FGVC [53], and CIFAR-100 [48]. ■ **Subject-Driven Generation**: Following [54], we use the DreamBooth dataset. More detailed descriptions of the datasets can be found in Appendix F.1, F.2, F.3.

**Pre-Trained Models**   We focus on a selection of representative pretrained models, including RoBERTa (Base & Large) [1], LLAMA3-8B [3], ViT (Base & Large) [55] and SDXL [56].

### 3.1.2   Baselines

In the baseline evaluation, we adopt a range of representative and competitive fine-tuning methods, categorized into three groups: Full Fine-Tuning, PEFT methods, and LoRA variants. The PEFT methods we use include BitFit [20], Adapter$^H$ [32], Adapter$^D$ [58], Adapter$^P$ [59], and LoRA [16]. For the LoRA variants, we consider AdaLoRA [26], DyLoRA [60], FourierFT [61], LoRA-drop [62], DoRA [63], MoSLoRA [28], PiSSA [64] and MiLoRA [57].

### 3.2   AuroRA Achieves Efficiency in NLP Tasks ($\mathcal{RQ}1$)

To answer $\mathcal{RQ}1$, we design two tasks: Natural Language Understanding (NLU) and Commonsense Reasoning. In the NLU task, we select RoBERTa-Base and RoBERTa-Large [1] as pretrained models and compare AuroRA with **ten** other widely-used fine-tuning methods across all six datasets of the GLUE benchmark [39]. The results of this extensive comparison are shown in Table 1, with additional hyperparameter configuration details provided in Appendix G.1. Following [16], we fine-tune only the query and value weights of each transformer block, while fully fine-tuning the classification head. For the commonsense reasoning task, we select LLaMA3-8B [3] as the base model and compare AuroRA with LoRA and two other LoRA variants (PiSSA [64] and MiLoRA [57]). The results are shown in Table 2. The relevant hyperparameters are listed in Appendix G.2. Our observations can be summarized as follows:

Table 3: Fine-tuning results with ViT Base and Large models on different image classification datasets. We report the accuracy (%) after 10 epochs. Avg. represents the average accuracy across all datasets for each method. * indicates numbers taken from [61]. The best results are highlighted in **bold**, and the runners-up are underlined (excluding full fine-tuning).

| Model | Method | Params. | OxfordPets | StanfordCars | CIFAR10 | DTD | EuroSAT | FGVC | RESISC45 | CIFAR100 | Avg. |
|---|---|---|---|---|---|---|---|---|---|---|---|
| ViT-Base | Full Fine-Tuning* | 85.8M | 93.1 | 79.8 | 98.9 | 77.7 | 99.1 | 54.8 | 96.1 | 92.4 | 86.5 |
| | Linear Probing* | - | 90.3$_{\downarrow 2.8}$ | 25.8$_{\downarrow 54.0}$ | 96.4$_{\downarrow 2.5}$ | 69.8$_{\downarrow 7.9}$ | 88.7$_{\downarrow 10.4}$ | 17.4$_{\downarrow 37.4}$ | 74.2$_{\downarrow 21.9}$ | 84.3$_{\downarrow 8.1}$ | 68.4$_{\downarrow 18.1}$ |
| | LoRA* | 581K | 93.2$_{\uparrow 0.1}$ | 45.4$_{\downarrow 34.4}$ | **98.8**$_{\downarrow 0.1}$ | 75.0$_{\downarrow 2.7}$ | 98.4$_{\downarrow 0.7}$ | 25.2$_{\downarrow 29.6}$ | 92.7$_{\downarrow 3.4}$ | **92.0**$_{\downarrow 0.4}$ | 77.6$_{\downarrow 8.9}$ |
| | FourierFT* | 239K | 93.1$_{\uparrow 0.0}$ | 56.4$_{\downarrow 23.4}$ | 98.7$_{\downarrow 0.2}$ | 77.3$_{\uparrow 0.4}$ | **98.8**$_{\downarrow 0.3}$ | 32.4$_{\downarrow 22.4}$ | **94.3**$_{\downarrow 1.8}$ | 91.5$_{\downarrow 0.9}$ | 80.3$_{\downarrow 6.2}$ |
| | AuroRA | 74K | **93.9**$_{\uparrow 0.8}$ | **75.7**$_{\downarrow 4.1}$ | **98.8**$_{\downarrow 0.1}$ | **79.6**$_{\uparrow 1.9}$ | **98.8**$_{\downarrow 0.1}$ | **48.2**$_{\downarrow 6.6}$ | 93.6$_{\downarrow 2.5}$ | **92.0**$_{\downarrow 0.4}$ | **85.1**$_{\downarrow 1.4}$ |
| ViT-Large | Full Fine-Tuning* | 303.3M | 94.4 | 88.9 | 99.2 | 81.8 | 99.0 | 68.3 | 96.4 | 93.6 | 90.2 |
| | Linear Probing* | - | 91.1$_{\downarrow 3.3}$ | 37.9$_{\downarrow 51.0}$ | 97.8$_{\downarrow 1.4}$ | 73.3$_{\downarrow 8.5}$ | 92.6$_{\downarrow 6.4}$ | 24.6$_{\downarrow 43.7}$ | 82.0$_{\downarrow 14.4}$ | 84.3$_{\downarrow 9.3}$ | 73.0$_{\downarrow 17.2}$ |
| | LoRA* | 1.57M | 94.8$_{\downarrow 0.4}$ | 73.3$_{\downarrow 15.6}$ | **99.1**$_{\downarrow 0.1}$ | 81.8$_{\uparrow 0.0}$ | 98.6$_{\downarrow 0.4}$ | 42.3$_{\downarrow 26.0}$ | 94.7$_{\downarrow 1.7}$ | **94.9**$_{\uparrow 1.3}$ | 84.9$_{\downarrow 5.3}$ |
| | FourierFT* | 480K | 94.8$_{\downarrow 0.4}$ | 79.1$_{\downarrow 9.8}$ | **99.1**$_{\downarrow 0.1}$ | 81.9$_{\uparrow 0.1}$ | 98.7$_{\downarrow 0.3}$ | 51.3$_{\downarrow 17.0}$ | **95.2**$_{\downarrow 1.2}$ | 93.4$_{\downarrow 0.2}$ | 86.7$_{\downarrow 3.5}$ |
| | AuroRA | 197K | **94.9**$_{\uparrow 0.5}$ | **82.5**$_{\downarrow 6.4}$ | **99.1**$_{\downarrow 0.1}$ | **82.1**$_{\uparrow 0.3}$ | **98.9**$_{\downarrow 0.1}$ | **59.8**$_{\downarrow 8.5}$ | 94.9$_{\downarrow 1.5}$ | 93.3$_{\downarrow 0.3}$ | **88.2**$_{\downarrow 2.0}$ |

**Obs. ❶ `AuroRA` demonstrates strong efficiency in NLP tasks.** It is evident that `AuroRA` outperforms the baseline across all datasets and pretrained models in both tasks. Compared to Full Fine-Tuning, `AuroRA` achieves a performance improvement ranging from $0.1\% \sim 8.3\%$ while using only $0.04\% \sim 0.06\%$ of the total parameters. Compared to PEFT baselines, including LoRA, `AuroRA` achieves a performance improvement of up to $24.7\%$ and an average improvement of $1.25\% \sim 10.88\%$, using only $6.25\% \sim 25\%$ of the parameters. Specifically, in the commonsense reasoning task using LLaMA3-8B as the pretrained model, `AuroRA` achieves a significant $10.7\%$ performance boost on ARC-C with just $6.25\%$ of LoRA's parameter budget. In the NLU task, although `AuroRA` uses more parameters than FourierFT, it demonstrates significant performance gains across all pretrained models and datasets. For instance, using RoBERTa-Base, `AuroRA` improves performance by $7.7\%$ on RTE.

**Obs. ❷ `AuroRA` can be scaled up to fine-tune large pretrained models.** `AuroRA` scales effectively to fine-tuning larger pretrained models. In the NLU task, when the pretrained model changes to RoBERTa-Large from Base, nearly all PEFT methods show a performance drop compared to Full Fine-Tuning, with the largest decrease reaching $3.7\%$. In contrast, `AuroRA` still achieves performance improvements of $0.1\% \sim 3.8\%$ across all datasets. In the commonsense reasoning task, when the model size increases to 8B, `AuroRA` continues to outperform LoRA by $2.4\% \sim 10.7\%$.

### 3.3 `AuroRA` Achieves Efficiency in CV Tasks ($\mathcal{RQ}2$)

To answer $\mathcal{RQ}2$, we design two tasks: Image Classification and Subject-Driven Image Generation. In the image classification task, following [61], we select ViT-Base and ViT-Large [55], two popular CV foundation models, which are pretrained on the ImageNet-21K [65] dataset. We then compare `AuroRA` with Full Fine-Tuning, Linear Probing (fine-tuning only the classification head), LoRA, and FourierFT. The results are presented in Table 3, with more implementation details available in Appendix G.3. In the subject-driven image generation task [54], following [61] and [28], we use the SDXL model [56] as our backbone, and then fine-tune it using both LoRA and `AuroRA`. The objective is to generate images based on specified prompts for a particular subject, which is defined using a set of reference images. Initially, we fine-tune a text-to-image model by pairing the input images with text prompts that include a unique identifier (e.g., "A photo of a [V] dog"). Subsequently, the model can generate images corresponding to other prompts that incorporate the same unique identifier, thereby producing images of the defined subject. The results are presented in Figure 4, and more generated cases are in Appendix H. Our observations can be summarized as follows:

**Obs. ❸ `AuroRA` achieves the best performance, excluding Full Fine-Tuning, with the least number of parameters.** It is evident that `AuroRA` outperforms all other PEFT baseline methods across all eight datasets with the lowest parameter count ($12.7\%$ of LoRA and $31.0\%$ of FourierFT) when using both the Base and Large models. Compared to Full Fine-Tuning, `AuroRA` uses only $0.086\%$ of the parameters and achieves a performance improvement of $0.4\% \sim 2.4\%$ on some datasets, with only a $1.6\% \sim 2.2\%$ gap in average performance. When using ViT-Base on STANFORDCARS, other baselines show a significant performance drop of $29.3\% \sim 67.7\%$ compared to Full Fine-Tuning. In contrast, `AuroRA` only experiences a moderate drop of $5.1\%$. Compared to PEFT methods, `AuroRA` achieves an average performance improvement of $1.73\% \sim 9.66\%$.

**Obs. ❹ `AuroRA` demonstrates stronger adaptability in the text-to-image domain.** We observe that in the subject-driven image generation task, `AuroRA` aligns better with the environment specified in the prompt. Specifically, when given the prompt "A [V] bear plushie on top of green grass with

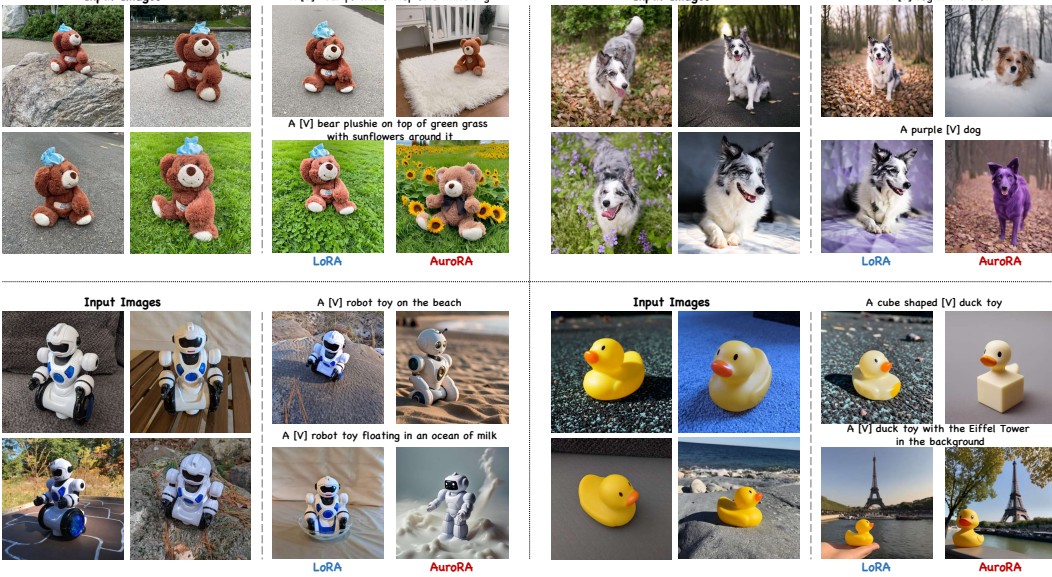

Figure 4: Results of LoRA and `AuroRA` in the subject-driven image generation task. `AuroRA` aligns better with the prompt.

sunflowers around it", LoRA generates an environment with only green grass but no sunflowers. In contrast, `AuroRA` successfully generates green grass with sunflowers.

## 3.4 Study

**Ablation Study ($\mathcal{RQ}$3)** To evaluate the contribution of different modules in `AuroRA`, we introduce two variants: (1) `AuroRA` w/o $\mathcal{F}$, and (2) `AuroRA` w/o $\mathcal{L}$, which correspond to the removal of the fixed and learnable nonlinearity in `AuroRA`, respectively. We compare these two variants with `AuroRA` by fine-tuning ViT-Base on OXFORDPETS, CIFAR10, DTD, and EUROSAT in the image classification task. From Table 4, we observe that: ❶ removing any component results in a performance drop for `AuroRA`; ❷ `AuroRA` w/o $\mathcal{L}$ consistently underperforms across all datasets, indicating that the learnable nonlinearity plays a more crucial role in the success of our method. Specifically, the learnable nonlinearity enables fine fitting, while the fixed nonlinearity contributes to coarse fitting.

Table 4: Comparison of different settings.

| Setting | OxfordPets | CIFAR10 | DTD | EuroSAT |
|---|---|---|---|---|
| `AuroRA` | 93.9 | 98.8 | 79.6 | 98.8 |
| `AuroRA` w/o $\mathcal{F}$ | 93.3 $_{\downarrow 0.6}$ | 98.4 $_{\downarrow 0.4}$ | 78.9 $_{\downarrow 0.7}$ | 98.3 $_{\downarrow 0.5}$ |
| `AuroRA` w/o $\mathcal{L}$ | 93.1 $_{\downarrow 0.8}$ | 98.2 $_{\downarrow 0.6}$ | 77.8 $_{\downarrow 1.8}$ | 98.0 $_{\downarrow 0.8}$ |

Table 5: Comparison of different activation functions.

| Setting | StanfordCars | FGVC | RESISC45 | CIFAR100 |
|---|---|---|---|---|
| `AuroRA` | 75.7 | 48.2 | 93.6 | 92.0 |
| `AuroRA`-lr | 75.6 $_{\downarrow 0.1}$ | 47.8 $_{\downarrow 0.4}$ | 93.4 $_{\downarrow 0.2}$ | 91.9 $_{\downarrow 0.1}$ |
| `AuroRA`-sm | 75.2 $_{\downarrow 0.5}$ | 47.7 $_{\downarrow 0.5}$ | 92.9 $_{\downarrow 0.7}$ | 91.7 $_{\downarrow 0.3}$ |

**Effect of Activation Function ($\mathcal{RQ}$4)** We investigate the impact of the choice of activation function in the fixed nonlinearity on `AuroRA`'s performance. Specifically, we introduce two variants: (1) `AuroRA`-lr, and (2) `AuroRA`-sm, which correspond to replacing the activation function in the fixed nonlinearity ($\tanh$) with LeakyReLU and Sigmoid, respectively. We compare these variants with `AuroRA` by fine-tuning the ViT-Base model on STANFORDCARS, FGVC, RESISC45, and CIFAR100 in the image classification task. From Table 5, we observe that Sigmoid results in the lowest performance, while $\tanh$ achieves the highest performance. Therefore, we choose $\tanh$ as the activation function for fixed nonlinearity in all our experiments.

**Sensitivity to Rank & Comparison with Linear LoRA Variants ($\mathcal{RQ}$5)** To further investigate the impact of introducing nonlinearity, we examine its sensitivity to rank and compare it with the linear LoRA variant under identical experimental settings. Specifically, we select LLaMA3-8B as the pretrained model and fine-tune it using `AuroRA`, MoSLoRA, and LoRA, varying the rank among $\{2, 4, 8, 16\}$. We evaluate their performance across four datasets. From Figure 5, we observe the following: ❶ the introduction of nonlinearity results in smaller performance fluctuations as the rank varies, i.e., more robustness to rank; ❷ `AuroRA` consistently outperforms across almost all rank settings and datasets, indicating that incorporating nonlinearity further enhances the model's expressiveness compared to linear approaches.

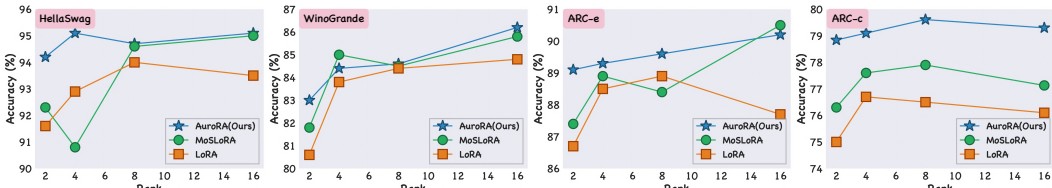

Figure 5: Performance comparison of different methods with varying ranks. We use LLaMA 3-8B as the pretrained model and fine-tune it using `AuroRA`, MoSLoRA, and LoRA methods on the HELLASWAG, WINOGRANDE, ARC-E and ARC-C datasets, with ranks $\{2, 4, 8, 16\}$.

## 4 Related Work

### 4.1 Parameter-Efficient Fine-Tuning

Parameter-Efficient Fine-Tuning (PEFT) has emerged as a pivotal strategy for addressing the computational challenges associated with fine-tuning large-scale pretrained models. PEFT methodologies can be broadly categorized into: ❶ **Additive PEFT** approaches introduce new, trainable modules to a frozen base model [18, 66, 32, 13, 67, 68, 15]. Common strategies include adapter-based techniques, such as AdapterFusion [59] and Hyperformer [69]; prompt-based methods, like Prefix-tuning [17] and p-tuning v2 [70]. ❷ **Selective PEFT** methods optimize a chosen subset of a pretrained model's parameters while keeping the majority frozen [71, 72, 73, 74, 75, 76]. This selection is often achieved through unstructured masking based on criteria like parameter significance, as seen in FishMask [77] and Child-tuning [78], or via structured techniques that group parameters, such as Bitfit [20] and SPT [79]. ❸ **Reparameterized PEFT** techniques transform model weights into more efficient, often low-rank, representations during fine-tuning, without altering the core architecture for inference [80, 60, 81, 82, 83]. A prominent example is LoRA [16], which introduces low-rank matrices for updates. ❹ **Memory-Efficient PEFT** methods focus on reducing the memory footprint of fine-tuning by optimizing the training dynamics rather than the model architecture [84, 85, 86, 87]. A representative example is GaLore [87], which projects gradients into low-rank subspaces to lower optimizer-state memory while preserving full-parameter adaptability. ❺ **Hybrid PEFT** methods integrate multiple strategies from different PEFT categories to capitalize on their respective advantages [19, 88, 89]. For instance, NOAH [90] and AUTOPEFT [91], leverage neural architecture search to identify effective PEFT combinations for specific tasks. In this paper, we primarily focus on LoRA, a reparameterized PEFT method.

### 4.2 LoRA and its Variants

The core idea of LoRA [16] is to approximate weight updates using mergeable, low-rank matrix pathways. Its variants can be broadly categorized into several types: ❶ **Novel Branch Designs** primarily focus on remodeling or reformulating the original low-rank matrix approximation pathway, with notable examples including VeRA [80], FourierFT [61], PiSSA [64], and DoRA [92]. ❷ **Multi-Task Variants**, exemplified by MoELoRA [81], MoA [82], CA-LoRA [93], and HydraLoRA [94], are engineered to enhance cross-task generalization—particularly in scenarios such as multi-task learning, domain adaptation, and continual learning—often through the strategic employment of LoRA module mixtures or ensembles. ❸ **Linear Variants**, including AdaLoRA [26], SaLoRA [27], MoSLoRA [28], and FLoRA [29], typically augment the LoRA framework by incorporating an additional linear matrix between the two original low-rank factors, thereby bolstering information capture during the training phase. Beyond these, several nonlinear LoRA variants have recently emerged, including LoRAN [95], SineLoRA [96], LoDA [97], NEAT [98], and CoLA [99]. However, these recent nonlinear variants do not resolve inherent low-rank bottleneck in LoRA. In contrast, our method pairs nonlinearities with a focus on LoRA's fundamental structural limitations, achieving a superior balance between performance and parameter efficiency.

## 5 Conclusion

In this paper, we revisit LoRA from the perspective of linear mappings and introduce nonlinearity into LoRA by proposing `AuroRA`, an MLP-like structure. `AuroRA` incorporates an adaptive nonlinear layer that includes both fixed and learnable nonlinearities between the two low-rank matrices. `AuroRA` achieves a superior balance between performance and parameters across tasks in both the NLP and CV domains. We hope that `AuroRA` will inspire further exploration of nonlinear extensions to LoRA.

**Limitation** A potential limitation is that, due to limited computational resources, we do not evaluate performance on larger pretrained models in this study, leaving this exploration for future work.

**Broader Impact** As a novel nonlinear method, `AuroRA` is envisioned for broad future applications in key sectors such as healthcare and finance. It is anticipated to deliver more accurate and reliable services while significantly reducing resource consumption, thereby better serving human society.

## Acknowledgments and Disclosure of Funding

This work is supported by the State Key Laboratory of General Artificial Intelligence; and the National Natural Science Foundation of China (Grant No. 62276006).

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

# A Notations

We summarize the notations used throughout the manuscript in Table 6.

Table 6: Notations commonly used in the `AuroRA` method.

| Notation | Definition |
|---|---|
| $\mathcal{W}_0 \in \mathbb{R}^{d_{\text{out}} \times d_{\text{in}}}$ | Pretrained weight matrix |
| $\Delta\mathcal{W} \in \mathbb{R}^{d_{\text{out}} \times d_{\text{in}}}$ | Weight update matrix |
| $\mathbf{A} \in \mathbb{R}^{\widetilde{r} \times d_{\text{in}}}$ | Downward projector (low-rank matrix) |
| $\mathbf{B} \in \mathbb{R}^{d_{\text{out}} \times \widetilde{r}}$ | Upward projector (low-rank matrix) |
| $r$ | Original LoRA rank ($r \ll \min(d_{\text{in}}, d_{\text{out}})$) |
| $\widetilde{r}$ | Compressed hidden dimension ($\widetilde{r} \ll r$) |
| $\mathbf{x} \in \mathbb{R}^{d_{\text{in}}}$ | Input vector |
| $\mathbf{h} \in \mathbb{R}^{d_{\text{out}}}$ | Output vector |
| $\sigma(\cdot)$ | Adaptive Nonlinear Layer (ANL) mapping $\mathbb{R}^{\widetilde{r}} \to \mathbb{R}^{\widetilde{r}}$ |
| $\mathcal{P}_{\text{down}}$ | Projection onto $\widetilde{r}$-dim hidden space (matrix $\mathbf{A}$) |
| $\mathcal{P}_{\text{self}}$ | Self-projection in hidden space (matrix $\mathbf{H} \in \mathbb{R}^{\widetilde{r} \times \widetilde{r}}$) |
| $\mathcal{P}_{\text{up}}$ | Projection back to $d_{\text{out}}$ (matrix $\mathbf{B}$) |
| $\mathcal{F}(\cdot)$ | Fixed nonlinearity (e.g. $\tanh$) |
| $\mathcal{L}(\cdot)$ | Learnable nonlinearity (B-spline based) |
| $\mathbf{w}_s \in \mathbb{R}^{\widetilde{r}}$ | Spline weight vector in $\mathcal{L}$ |
| $\mathbf{s}(\mathbf{Z})$ | Spline basis functions applied to each component of $\mathbf{Z}$ |
| $T$ | Number of training epochs |

# B Complete Process

In this section, we first provide further details on both the static weight merging operation performed after the training phase and the actual process at inference time. We then offer empirical validation for our approach. Finally, the complete algorithm workflow of our `AuroRA` is presented in Algo. 1.

## B.1 Weights Merging & Inference Phase

During the training phase, to enhance training flexibility, `AuroRA` utilizes a *dynamic*, input-dependent update mechanism formulated as $\mathbf{B} \cdot \sigma(\mathbf{A}\mathbf{x})$. After the training phase, once all learnable parameters are fixed and considered optimized, `AuroRA` transitions to a *static* form for the inference stage. This static form, given by $\Delta\mathcal{W} = \mathbf{B} \cdot \sigma(\mathbf{A})$, facilitates the seamless integration of `AuroRA` into the pre-trained weight $\mathcal{W}_0$, consistent with standard LoRA. Therefore, after the weights are merged, the effective forward propagation process at inference time is formally given by:

$$\mathbf{h} = \mathcal{W}_0\mathbf{x} + \Delta\mathcal{W}\mathbf{x} = \mathcal{W}_0\mathbf{x} + \mathbf{B} \cdot \sigma(\mathbf{A})\mathbf{x}. \tag{9}$$

Such an approach eliminates any additional computational overhead during inference, while concurrently preserving superior performance.

**Why is This Strategy Effective?** To validate our strategy of *dynamic* training combined with *static* inference, we empirically compare it against both *fully dynamic* and *fully static* approaches. For this purpose, we introduce two comparative variants: (1) `AuroRA`-$\mathcal{D}$, which maintains dynamic processing throughout both training and inference (i.e., its forward pass is consistently $\mathbf{h} = \mathcal{W}_0\mathbf{x} + \mathbf{B} \cdot \sigma(\mathbf{A}\mathbf{x})$), and (2) `AuroRA`-$\mathcal{S}$, which consistently employs a static form for both phases (i.e., $\mathbf{h} = \mathcal{W}_0\mathbf{x} + \mathbf{B} \cdot \sigma(\mathbf{A})\mathbf{x}$). We evaluate `AuroRA`, `AuroRA`-$\mathcal{D}$, and `AuroRA`-$\mathcal{S}$ by fine-tuning LLaMA3-8B for Commonsense Reasoning on datasets including ARC-E, OBQA, SIQA, and ARC-C, and record both accuracy and total training and inference time. From Table 7, we observe that: ❶ `AuroRA` exhibits nearly identical performance to `AuroRA`-$\mathcal{D}$ and significantly outperforms `AuroRA`-$\mathcal{S}$ in

Table 7: Comparison of accuracy and total time consumption for different settings on eight Commonsense Reasoning datasets, using LLaMA3-8B as pre-trained model.

| Setting | Time | BoolQ | PIQA | SIQA | HellaSwag | WinoGrande | ARC-e | ARC-c | OBQA | Avg. |
|---|---|---|---|---|---|---|---|---|---|---|
| LoRA | 15.05 h | 70.8 | 85.2 | 79.9 | 91.7 | 84.3 | 84.2 | 71.2 | 79.0 | 80.8 |
| AuroRA-$\mathcal{S}$ | 15.18 h | 71.4 | 86.9 | 78.1 | 93.5 | 81.7 | 88.5 | 78.1 | 83.9 | 82.8 |
| AuroRA-$\mathcal{D}$ | 15.60 h | 72.6 | 87.5 | 79.2 | 94.3 | 83.0 | 89.5 | 78.9 | 85.0 | 83.8 |
| AuroRA | 15.28 h | 72.5 | 87.4 | 79.0 | 94.2 | 83.0 | 89.3 | 78.8 | 84.8 | 83.6 |

practice; ❷ AuroRA achieves a runtime comparable to that of AuroRA-$\mathcal{S}$ and is markedly faster than AuroRA-$\mathcal{D}$. Therefore, our practical implementation adopts this dynamic training with static inference strategy, which can be seamlessly merged into pre-trained weights after the training phase (consistent with standard LoRA), and thereby also achieves an effective trade-off between performance and computational cost.

### B.2 Algorithm Workflow

The algorithm framework is presented in Algo. 1.

---

**Algorithm 1:** Algorithm workflow of AuroRA

---

**Input** : Pretrained weight $\mathcal{W}_0$, low-rank factors $\mathbf{A} \in \mathbb{R}^{\tilde{r} \times d}$ and $\mathbf{B} \in \mathbb{R}^{d \times \tilde{r}}$, ANL parameters, training data $\{(\mathbf{x}_i, y_i)\}_{i=1}^N$, number of epochs $T$

/* Training Phase (dynamic update: $\mathbf{B}\,\sigma(\mathbf{Ax})$) */
**for** epoch $t \leftarrow 1$ **to** $T$ **do**
    **for** *each minibatch* $\{\mathbf{x}, y\}$ *in training data* **do**
        /* Forward pass with ANL on input */
        $\mathbf{h} \leftarrow \mathcal{W}_0\,\mathbf{x} + \mathbf{B} \cdot \sigma(\mathbf{A}\,\mathbf{x})$ ;    ▷ Eq. 4
        Compute loss $\mathcal{L}(\mathbf{h}, y)$
        /* Backpropagate through $\mathbf{A}, \mathbf{B}$, and ANL parameters */
        Backpropagate and update $\{\mathbf{A}, \mathbf{B}, \mathrm{ANL}\}$
    **end**
**end**

/* Inference Preparation (static merge: $\mathbf{B}\,\sigma(\mathbf{A})$) */
**Function** *MergeWeights()*:
    /* Compute element-wise ANL on matrix $\mathbf{A}$ */
    $\widetilde{\mathbf{A}} \leftarrow \sigma(\mathbf{A})$
    /* Form the effective weight update */
    $\Delta\mathcal{W} \leftarrow \mathbf{B}\,\widetilde{\mathbf{A}}$ ;    ▷ Eq. 3
    /* Merge into pretrained weights */
    $\mathcal{W} \leftarrow \mathcal{W}_0 + \Delta\mathcal{W}$
    **return** $\mathcal{W}$

/* Inference Phase (static forward: no extra ANL) */
**for** *each test sample* $\mathbf{x}$ **do**
    $\mathcal{W} \leftarrow \mathrm{MergeWeights}()$
    $\mathbf{h} \leftarrow \mathcal{W}\,\mathbf{x}$ ;    ▷ Eq. 9
    /* Use $\mathbf{h}$ for downstream prediction */
**end**

---

## C  Intuitive Case

To intuitively and concisely illustrate the impact of nonlinear mapping on the matrices, we first consider a simple scenario where $\mathbf{A} \in \mathbb{R}^{1 \times 2}$ and $\mathbf{B} \in \mathbb{R}^{2 \times 1}$:

$$\mathbf{A} = \begin{bmatrix} a_1 & a_2 \end{bmatrix}, \quad \mathbf{B} = \begin{bmatrix} b_1 \\ b_2 \end{bmatrix} \tag{10}$$

We then introduce the LeakyReLU activation function as the nonlinear mapping between the two low-rank matrices. Depending on the elements of matrix $\mathbf{A}$, the resulting weight update comprises the following four matrix structures:

$$\Delta\mathcal{W} = \begin{bmatrix} b_1 a_1 & b_2 a_1 \\ b_1 a_2 & b_2 a_2 \end{bmatrix}, \qquad \text{if } a_1 > 0 \text{ and } a_2 > 0,$$

$$\begin{bmatrix} b_1(\alpha a_1) & b_2(\alpha a_1) \\ b_1 a_2 & b_2 a_2 \end{bmatrix}, \quad \text{if } a_1 \le 0 \text{ and } a_2 > 0,$$

$$\begin{bmatrix} b_1 a_1 & b_2 a_1 \\ b_1(\alpha a_2) & b_2(\alpha a_2) \end{bmatrix}, \quad \text{if } a_1 > 0 \text{ and } a_2 \le 0, \tag{11}$$

$$\begin{bmatrix} b_1(\alpha a_1) & b_2(\alpha a_1) \\ b_1(\alpha a_2) & b_2(\alpha a_2) \end{bmatrix}, \quad \text{if } a_1 \le 0 \text{ and } a_2 \le 0.$$

where $\alpha$ is a hyperparameter of LeakyReLU, usually a positive number less than 1. Meanwhile, LoRA can only produce:

$$\Delta\mathcal{W} = \begin{bmatrix} b_1 a_1 & b_2 a_1 \\ b_1 a_2 & b_2 a_2 \end{bmatrix} \tag{12}$$

Under the LeakyReLU activation function, each negative component of $\mathbf{A}$ is scaled by the factor $\alpha$, while positive components remain unchanged. This piecewise linear mapping disrupts the uniformity of low-rank multiplication, causing the final weight update $\Delta\mathcal{W}$ to depend not only on the product of $\mathbf{A}$ and $\mathbf{B}$ but also on the local behavior of each element in $\mathbf{A}$. Consequently, even under tight rank constraints, the model benefits from a richer set of possible weight updates, enhancing its adaptability to varying input distributions.

## D    Proof of Proposition 2.1

In this appendix, we provide the complete theoretical analysis and proof of Proposition 2.1. We first restate the problem setup and then present the necessary lemmas, followed by the main proof.

**Definition D.1** (Best Linear Rank-$r$ Error). For $M \in \mathbb{R}^{d_{\text{out}} \times d_{\text{in}}}$, define

$$\varepsilon_r(M) := \inf_{U \in \mathbb{R}^{d_{\text{out}} \times r}, V \in \mathbb{R}^{r \times d_{\text{in}}}} \| M - UV \|.$$

If $\operatorname{rank}(M) > r$, then $\varepsilon_r(M) > 0$ [100].

**Assumption D.2** (Bounded Input Domain). We assume $\mathbf{x} \in \mathcal{X} \subset \mathbb{R}^{d_{\text{in}}}$ satisfies $\|\mathbf{x}\| \le X_{\max}$. Then for $A \in \mathbb{R}^{r \times d_{\text{in}}}$ with $\|A\| \le A_{\max}$, the vector $\mathbf{z} = A\mathbf{x}$ remains in a bounded set $\Omega \subset \mathbb{R}^r$ (compact).

**Definition D.3** (Nonlinear Low-Rank Update). Let

$$M_{\text{nonlinear}}(\mathbf{x}) = B\,\sigma(A\mathbf{x}),$$

with $A \in \mathbb{R}^{r \times d_{\text{in}}}, B \in \mathbb{R}^{d_{\text{out}} \times r}$. The map $\sigma : \mathbb{R}^r \to \mathbb{R}^r$ is given by

$$\sigma(\mathbf{z}) = \mathcal{F}(\mathbf{z}) + \mathbf{w}_s \cdot \mathbf{s}(\mathbf{z}),$$

where $\mathcal{F}$ is a fixed bounded function (e.g. $\tanh$-based) and $\mathbf{s}(\mathbf{z})$ denotes B-spline basis functions in $\mathbb{R}^r$.

**Lemma D.4** (Piecewise Polynomial Approximation). *Consider $f : \Omega \to \mathbb{R}^m$ with $f \in C^k(\Omega)$ on a bounded domain $\Omega \subset \mathbb{R}^r$. Let $\Delta > 0$ be the subdivision size in each coordinate axis for constructing a tensor-product B-spline. Then there exists a B-spline $g(\mathbf{z})$ such that*

$$\sup_{\mathbf{z} \in \Omega} \| f(\mathbf{z}) - g(\mathbf{z}) \| \le C_f(\Delta)^k,$$

*where $C_f > 0$ is a constant depending on $f$'s $k$-th order partial derivatives and the geometry of $\Omega$ [34].*

**Lemma D.5** (Combining Fixed and Learnable Nonlinearities). *Let $\Omega \subset \mathbb{R}^r$ be compact. Assume $\mathcal{F} : \Omega \to \mathbb{R}^r$ is fixed, bounded, and $C^1$, and let $h(\mathbf{z}) \in C^k(\Omega)$ be the target. Define*

$$\sigma(\mathbf{z}) = \mathcal{F}(\mathbf{z}) + \mathbf{w}_s \cdot \mathbf{s}(\mathbf{z}),$$

*where $\mathbf{s}(\mathbf{z})$ is a B-spline basis. Then, for any $\epsilon > 0$, one can choose $\Delta > 0$ and $\mathbf{w}_s$ such that*

$$\sup_{\mathbf{z} \in \Omega} \left\| h(\mathbf{z}) - \left[ \mathcal{F}(\mathbf{z}) + \mathbf{w}_s \cdot \mathbf{s}(\mathbf{z}) \right] \right\| \leq \epsilon.$$

*Furthermore, the error decays like $O\big((\Delta)^k\big)$ as $\Delta \to 0$.*

*Proof of Lemma D.5.* Let $r(\mathbf{z}) = h(\mathbf{z}) - \mathcal{F}(\mathbf{z})$. Since $h \in C^k(\Omega)$ and $\mathcal{F}$ is fixed and $C^1$, $r(\mathbf{z})$ remains $C^k$. Applying Lemma D.4 to $r(\mathbf{z})$ yields a B-spline $g(\mathbf{z})$ with $\|r(\mathbf{z}) - g(\mathbf{z})\| \leq C(\Delta)^k$. Hence $\|h(\mathbf{z}) - [\mathcal{F}(\mathbf{z}) + g(\mathbf{z})]\| \leq C(\Delta)^k$, completing the proof. $\qquad \square$

We restate Proposition 2.1 here for completeness:

**Proposition 2.1** (Lower Approximation Error) *Let $M \in \mathbb{R}^{d_{\text{out}} \times d_{\text{in}}}$ with $\text{rank}(M) > r$. Then*

$$\varepsilon_r(M) := \inf_{U, V} \|M - UV\| > 0.$$

*Under Definition D.3, there exist $A^*, B^*$ and a B-spline parameter set $(\mathbf{w}_s^*)$ such that*

$$\| M - M_{\text{nonlinear}} \| \leq c\, \varepsilon_r(M), \quad 0 < c < 1.$$

*Proof.* Let $M^* = U^* V^*$ be the best linear rank-$r$ approximation of $M$, so $\|M - M^*\| = \varepsilon_r(M)$. Denote the residual $R = M - M^*$, and we have $\|R\| = \varepsilon_r(M)$.

By Assumption D.2, for $\|\mathbf{x}\| \leq X_{\max}$, let $\mathbf{z} = A^* \mathbf{x} \in \Omega \subset \mathbb{R}^r$, with $\|A^*\| \leq A_{\max}$. Thus $\mathbf{z}$ lies in a compact $\Omega$. Consider $R(\mathbf{x})$ as a function $h(\mathbf{z}) = R(\mathbf{x})$. Since $R$ is linear (hence $C^\infty$), $h(\mathbf{z})$ is at least $C^1$ in $\mathbf{z}$.

From Lemma D.5, there is a B-spline $\mathbf{w}_s^* \cdot \mathbf{s}(\mathbf{z})$ approximating $h(\mathbf{z}) - \mathcal{F}(\mathbf{z})$ within $\gamma \|R\|$ for some $0 < \gamma < 1$. Define

$$\widehat{M}(\mathbf{x}) := M^*(\mathbf{x}) + B^* \Big[ \mathcal{F}(A^* \mathbf{x}) + \mathbf{w}_s^* \cdot \mathbf{s}(A^* \mathbf{x}) \Big].$$

Then

$$\|\widehat{M} - M\| = \left\| \big(M^* + B^*[\dots]\big) - (M^* + R) \right\| = \left\| B^* \big[ \mathcal{F}(\cdot) + \mathbf{w}_s^* \cdot \mathbf{s}(\cdot) \big] - R \right\| \leq \gamma \|R\| = \gamma\, \varepsilon_r(M).$$

Since $\gamma < 1$, we obtain $\|\widehat{M} - M\| < \varepsilon_r(M)$, which strictly improves upon the LoRA limit. Setting $\widehat{M} \equiv M_{\text{nonlinear}}$ completes the proof. $\qquad \square$

# E   Proof of Proposition 2.2

**Proposition 2.2.** *In the* `AuroRA`, *the use of the* $\tanh$ *activation function and B-spline basis functions results in bounded gradients with respect to both the inputs and the model parameters.*

The loss function $L$ for a single data point $(x, y)$ is defined as $L(x, y) = \frac{1}{2} \|f_{\texttt{AuroRA}}(x) - y\|^2$, where $y \in \mathbb{R}^{d_{\text{out}}}$ is the target output. We will compute and bound the gradients of the loss function with respect to $W_b$, $w_s$ and the input $x$.

**Lemma E.1.** *The gradients of the loss function with respect to $W_b$ is bounded.*

*Proof.* Compute the gradient $\frac{\partial L}{\partial W_b}$:

$$\frac{\partial L}{\partial W_b} = (f_{\texttt{AuroRA}}(x) - y)^\top B \cdot \frac{\partial \text{ANL}(A^\top x)}{\partial W_b}.$$

Compute $\frac{\partial \text{ANL}(z)}{\partial W_b}$:

$$\frac{\partial \text{ANL}(z)}{\partial W_b} = \frac{\partial \phi(W_b \phi(z))}{\partial W_b} = \text{diag}\left( \phi'(W_b \phi(z)) \right) \cdot \phi(z)^\top,$$

where $\phi'(u) = 1 - \tanh^2(u)$ is the derivative of Tanh, $\operatorname{diag}(v)$ denotes a diagonal matrix with vector $v$ on the diagonal. It is not difficult to deduce that $\phi(z) \in (-1, 1)$ since Tanh outputs are bounded, and $\phi'(u) \in (0, 1]$ because $1 - \tanh^2(u) \leq 1$. So $\frac{\partial \text{ANL}(z)}{\partial W_b}$ is bounded. Consequently, $\frac{\partial L}{\partial W_b}$ is bounded as it is a product of bounded terms. $\square$

**Lemma E.2.** *The gradients of the loss function with respect to $w_s$ is bounded.*

*Proof.* Compute the gradient $\frac{\partial L}{\partial w_s}$:

$$\frac{\partial L}{\partial w_s} = (f_{\texttt{AuroRA}}(x) - y)^\top B \cdot \frac{\partial \text{ANL}(A^\top x)}{\partial w_s}.$$

Compute $\frac{\partial \text{ANL}(z)}{\partial w_s}$:

$$\frac{\partial \text{ANL}(z)}{\partial w_s} = s(z),$$

since $\text{ANL}(z)$ is linear in $w_s$. B-spline basis functions $B(z_i)$ are smooth and have compact support, and the outputs of $B(z_i)$ are bounded. Therefore, $s(z)$ is bounded, and thus $\frac{\partial L}{\partial w_s}$ is bounded. $\square$

**Lemma E.3.** *The gradients of the loss function with respect to the input $x$ is bounded.*

*Proof.* Compute the gradient $\frac{\partial L}{\partial x}$:

$$\frac{\partial L}{\partial x} = (f_{\texttt{AuroRA}}(x) - y)^\top \left( W + B \cdot \frac{\partial \text{ANL}(A^\top x)}{\partial x} \right).$$

Compute $\frac{\partial \text{ANL}(A^\top x)}{\partial x}$:

$$\frac{\partial \text{ANL}(A^\top x)}{\partial x} = \frac{\partial \text{ANL}(z)}{\partial z} \cdot A^\top,$$

where $z = A^\top x$. Compute $\frac{\partial \text{ANL}(z)}{\partial z}$:

$$\frac{\partial \text{ANL}(z)}{\partial z} = \frac{\partial \phi(W_b \phi(z))}{\partial z} + w_s \cdot \frac{\partial s(z)}{\partial z}.$$

Compute $\frac{\partial \phi(W_b \phi(z))}{\partial z}$:

$$\frac{\partial \phi(W_b \phi(z))}{\partial z} = \operatorname{diag}\left( \phi'\left( W_b \phi(z) \right) \right) W_b \operatorname{diag}\left( \phi'(z) \right).$$

$\phi'(z)$ and $\phi'\left( W_b \phi(z) \right)$ are bounded in $(0, 1]$. Entries of $W_b$ are finite. So $\frac{\partial \phi(W_b \phi(z))}{\partial z}$ is bounded.

Compute $\frac{\partial s(z)}{\partial z}$:

$$\frac{\partial s(z)}{\partial z} = [B'(z_1), B'(z_2), \ldots, B'(z_r)]^\top.$$

Derivatives $B'(z_i)$ of B-spline functions are bounded due to their polynomial nature and compact support. So $\frac{\partial s(z)}{\partial z}$ is bounded. Therefore, $\frac{\partial \text{ANL}(z)}{\partial z}$ is bounded, leading to $\frac{\partial \text{ANL}(A^\top x)}{\partial x}$ being bounded. Consequently, $\frac{\partial L}{\partial x}$ is bounded. $\square$

# F  Dataset

## F.1  GLUE Benchmark

The GLUE (General Language Understanding Evaluation), as introduced in [39], is a widely adopted benchmark in the field of Natural Language Processing (NLP). GLUE encompasses a collection of eight diverse NLP tasks: MNLI (natural language inference), SST-2 (sentiment analysis), MRPC (paraphrase detection), CoLA (linguistic acceptability), QNLI (natural language inference), QQP (question answering), RTE (recognizing textual entailment), and STS-B (textual similarity). The statistical details of these datasets are summarized in Table 8.

Table 8: Detailed task descriptions and dataset statistics for the GLUE benchmark. STS-B is categorized as a regression task, while all other tasks involve single-sentence or sentence-pair classification.

| Corpus | Task | Metrics | # Labels | # Train | # Val | # Test | Domain |
|--------|------|---------|----------|---------|-------|--------|--------|
| | | Single-Sentence Tasks | | | | | |
| CoLA | Acceptability | Matthews Corr. | 2 | 8.55k | 1.04k | 1.06k | misc. |
| SST-2 | Sentiment | Accuracy | 2 | 67.3k | 872 | 1.82k | Movie reviews |
| | | Similarity and Paraphrase Tasks | | | | | |
| MRPC | Paraphrase | Accuracy/F1 | 2 | 3.67 | 408 | 1.73k | News |
| STS-B | Sentence similarity | Pearson/Spearman Corr. | 1 | 5.75k | 1.5k | 1.38k | misc. |
| QQP | Paraphrase | Accuracy/F1 | 2 | 364k | 40.4k | 391k | Social QA |
| | | Inference Tasks | | | | | |
| MNLI | NLI | Accuracy | 3 | 393k | 19.65k | 19.65k | misc. |
| QNLI | QA/NLI | Accuracy | 2 | 105k | 5.46k | 5.46k | Wikipedia |
| RTE | NLI | Accuracy | 2 | 2.49k | 277 | 3k | News & Wikipedia |

## F.2 Commonsense Reasoning

Following [19], we use eight datasets in Commonsense Reasoning task. (1) The BoolQ [40] dataset is a question-answering benchmark consisting of 15,942 examples, where the questions are naturally occurring and generated in unprompted and unconstrained settings, requiring yes/no answers. (2) The PIQA [41] dataset presents questions with two potential solutions, demanding physical commonsense reasoning to identify the correct answer. (3) The SIQA [42] dataset focuses on reasoning about human actions and their social implications. (4) The HellaSwag [43] dataset is designed for commonsense natural language inference (NLI) tasks, where each question includes a context and several potential endings, from which the correct continuation must be selected. (5) The WinoGrande [44] dataset is a fill-in-the-blank task with binary options, where the goal is to select the most plausible option for a given sentence requiring commonsense reasoning. (6) The ARC-c and (7) ARC-e [45] datasets refer to the Challenge and Easy sets, respectively, of the ARC dataset, which consists of multiple-choice science questions designed at a grade-school level, with the former being more challenging than the latter. (8) The OBQA [46] dataset focuses on questions that necessitate multi-step reasoning, integration of external common knowledge, and in-depth text comprehension. Statistical details are shown in Table 9.

Table 9: Details of datasets being evaluated in commonsense reasoning task.

| Dataset | #Train | #Test | Answer |
|---------|--------|-------|--------|
| BoolQ [40] | 9.4K | 3270 | Yes/No |
| PIQA [41] | 16.1K | 1830 | Option |
| SIQA [42] | 33.4K | 1954 | Option |
| HellaSwag [43] | 39.9K | 10042 | Option |
| WinoGrande [44] | 63.2K | 1267 | Option |
| ARC-e [45] | 1.1K | 2376 | Option |
| ARC-c [45] | 2.3K | 1172 | Option |
| OBQA [46] | 5.0K | 500 | Option |

## F.3 Image Classification

We show the details of the datasets in Image Classification task in Table 10.

## G  Hyperparameters

To ensure the reproducibility of our experimental results, we provide the detailed hyperparameter settings used in our experiments. In all of our experiments, to achieve a better balance between parameter count and performance, we set the hidden layer dimension (Rank $\widetilde{r}$) of `AuroRA` to 2. Correspondingly, we set the hyperparameter $\alpha$ of `AuroRA` to 4. Natural Language Understanding and

Table 10: Details of the datasets for the Image Classification task.

| Dataset | #Class | #Train | #Val | #Test | Rescaled resolution |
|---------|--------|--------|------|-------|---------------------|
| OxfordPets [47] | 37 | 3,312 | 368 | 3,669 | |
| StandfordCars [52] | 196 | 7,329 | 815 | 8,041 | |
| CIFAR10 [48] | 10 | 45,000 | 5,000 | 10,000 | |
| DTD [49] | 47 | 4,060 | 452 | 1,128 | $224 \times 224$ |
| EuroSAT [50] | 10 | 16,200 | 5,400 | 5,400 | |
| FGVC [53] | 100 | 3,000 | 334 | 3,333 | |
| RESISC45 [51] | 45 | 18,900 | 6,300 | 6,300 | |
| CIFAR100 [48] | 100 | 45,000 | 5,000 | 10,000 | |

image classification tasks run on four NVIDIA GeForce RTX 4090 (24GB) GPUs. Commonsense reasoning and subject-driven generation tasks run on NVIDIA L20 (48GB).

## G.1  Natural Language Understanding

We provide the hyperparameters used for the GLUE benchmark in natural language understanding experiments in Table 11. To facilitate reproducibility, we fix the random seed to 0. We tune the learning rate, while all other settings follow those used in LoRA [16] and FourierFT [61].

Table 11: Hyperparameter setup of `AuroRA` for the GLUE benchmark.

| Model | Hyperparameter | STS-B | RTE | MRPC | CoLA | SST-2 | QNLI |
|-------|----------------|-------|-----|------|------|-------|------|
| Both | Optimizer | | | | AdamW | | |
| | LR Schedule | | | | Linear | | |
| | Warmup Ratio | | | | 0.06 | | |
| | Rank $\widetilde{r}$ | | | | 2 | | |
| | $\alpha$ | | | | 4 | | |
| Base | Epochs | 30 | 80 | 30 | 90 | 30 | 80 |
| | Learning Rate | 6E-4 | 5E-4 | 8E-4 | 5E-3 | 8E-4 | 5E-3 |
| | Max Seq. Len | 512 | 512 | 512 | 512 | 512 | 512 |
| | Batch Size | 64 | 16 | 64 | 32 | 32 | 32 |
| Large | Epochs | 20 | 10 | 30 | 50 | 40 | 20 |
| | Learning Rate | 3E-4 | 4E-4 | 1E-3 | 5E-4 | 8E-4 | 4E-4 |
| | Max Seq. Len | 512 | 512 | 512 | 256 | 128 | 512 |
| | Batch Size | 16 | 32 | 16 | 16 | 16 | 16 |

## G.2  Commonsense Reasoning

We provide the detailed hyperparameters for fine-tuning LLaMA3-8B in the commonsense reasoning task in Table 12.

Table 12: Hyperparameter setup of `AuroRA` for Commonsense Reasoning.

| Hyperparameter | Commonsense Reasoning |
|----------------|----------------------|
| Rank $\widetilde{r}$ | 2 |
| $\alpha$ | 4 |
| Dropout | 0.05 |
| Batch Size | 16 |
| Optimizer | Adam W |
| Learning Rate | 3e-4 |
| Warmup Steps | 100 |
| Epochs | 3 |
| Target module | q,k,v,up,down |

## G.3 Image Classification

We provide the detailed hyperparameters for the image classification in Table 13. We tune the learning rate, while the weight decay value follows the settings used in FourierFT [61] without tuning.

Table 13: Hyperparameter setup of `AuroRA` for the image classification.

| Model | Hyperparameter | OxfordPets | StanfordCars | CIFAR10 | DTD | EuroSAT | FGVC | RESISC45 | CIFAR100 |
|---|---|---|---|---|---|---|---|---|---|
| Both | Optimizer
LR Schedule
Epochs
Rank $\tilde{r}$
$\alpha$ | | | | AdamW
Linear
10
2
4 | | | | |
| Base | Learning Rate (`AuroRA`)
Learning Rate (Head)
Weight Decay | 5e-3
5E-3
8E-4 | 1e-2
1e-2
4E-5 | 1e-2
3e-2
9E-5 | 1e-2
8E-3
7E-5 | 5e-3
8E-3
3E-4 | 1e-2
1e-2
7E-5 | 8e-3
1e-2
3E-4 | 8e-3
5e-3
1E-4 |
| Large | Learning Rate (`AuroRA`)
Learning Rate (Head)
Weight Decay | 5e-3
4e-3
8E-4 | 9e-3
8e-3
4E-5 | 8e-3
4e-2
9E-5 | 8e-3
9e-3
7E-5 | 4e-3
8e-3
3E-4 | 1.5e-2
1e-2
7E-5 | 7.5e-3
1.5e-2
3E-4 | 1.5e-2
5e-3
1E-4 |

# H   More Cases of Generated Images

In Figure 6, we present more results of subject-driven generation using both LoRA and `AuroRA`.

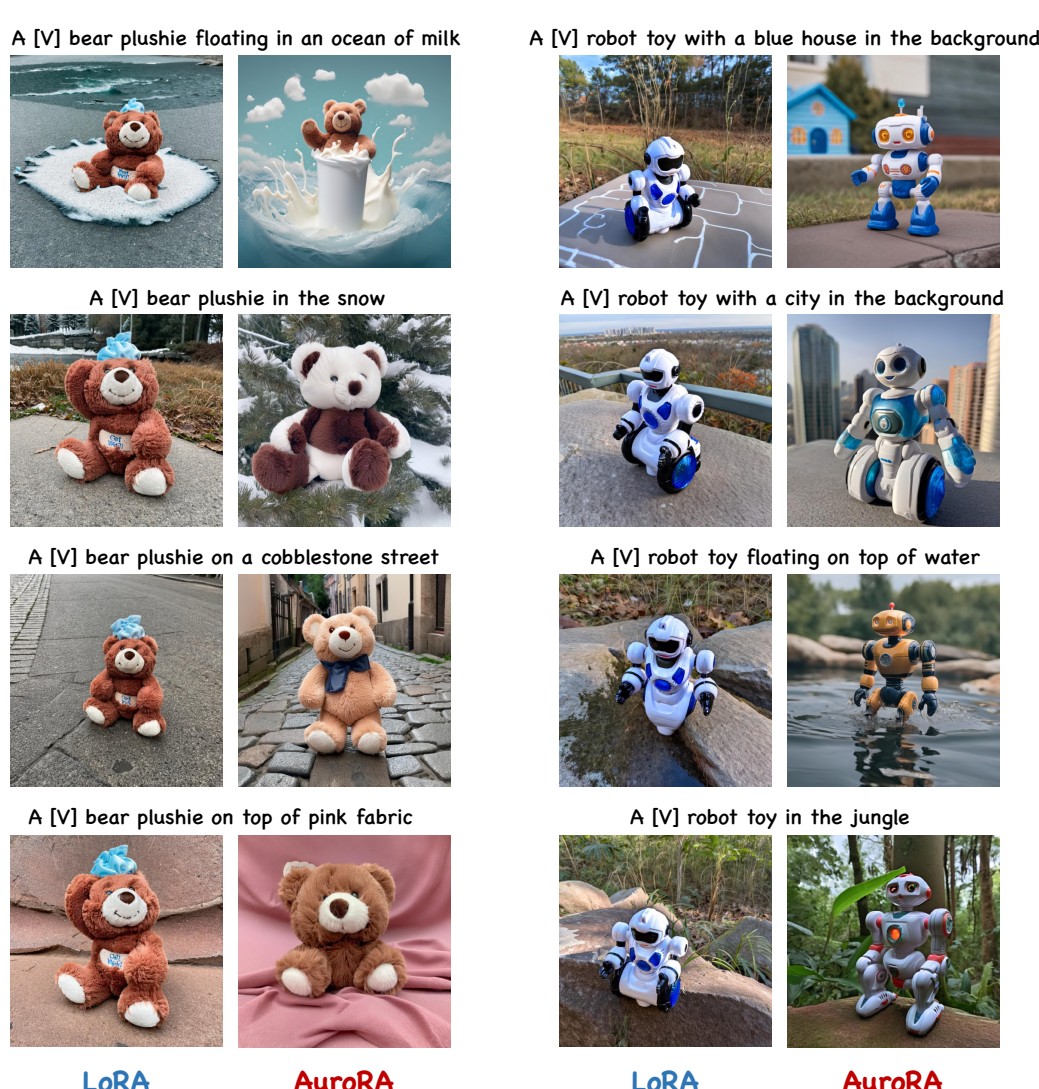

Figure 6: More generated images in the subject-driven generation task.

