# OpenReview forum: "AuroRA: Breaking Low-Rank Bottleneck of LoRA with Nonlinear Mapping"
_NeurIPS.cc/2025/Conference — NeurIPS 2025 spotlight_

### Official Review · Reviewer_4VGD · 2025-06-18

**Clarity:** 2
**Significance:** 3
**Originality:** 3
**Rating:** 5
**Confidence:** 4

**Summary:**

AuroRA is a modification of the standard LoRA PEFT architecture which introduces non-linearities into the low rank bottleneck to capture more complex subspace relationships with minimal additional parameters, enabling AuroRA to outperform linear techniques such as LoRA and MoSLoRA across NLP and CV tasks.

AuroRA introduces the "Adaptive Nonlinear Layer" (ANL) as the non-linearity used in the low rank bottleneck. The ANL is a combination of (1) fixed non-linearities with an intermediate 'self projection' similar to the subspace mixer of MoSLoRA which facilitates smoother fitting of subspace relationships, and (2) a learnable non-linearity with learnable splines similar to that of Kolmogorov–Arnold Networks (KANs) which enables tighter fitting of relationships within the low-rank subspace. These non-linearities enable AuroRA to explore much wider $\Delta w$ parameter spaces than classical PEFT methods, which the authors identify to be the primary weakness of LoRA and LoRA-like adaptors.

**Questions:**

Did the authors perform alpha tuning to determine the best LoRA, MoSLoRA and AuroRA hyperparameters for the comparison ablations, and if so what values were chosen?

Were static or dynamic weights utilised when collecting the inference results in the main body of the paper?

Can the authors justify how it is mathematically possible to linearize AuroRA considering its data-dependent activations and what the implications are considering it re-introduces a linear low-rank bottleneck.

**Ethical Concerns:**

["NO or VERY MINOR ethics concerns only"]

**Final Justification:**

The authors addressed all my major concerns, and I now believe AuroRA to be a strong and impactful contender in the world of PEFT techniques, especially due to the justification that AuroRA's non-linear components can be "baked in" to a linear approximation that fully allows weight merging with only a minor loss in performance which is something novel and highly advantageous over other PEFT methods that utilise non-linearities.

**Limitations:**

Yes

**Quality:**

3

**Strengths And Weaknesses:**

## Strengths
While the introduction of bottleneck MLPs for PEFT is not entirely new, their use as a replacement for LoRA-like weight adaptation is novel and shows incredibly promising results.

The authors perform a wide variety of ablations which support their design decisions and perform experiments with a selection of standard models and tasks that are typical to the NLP and CV domains. Despite AuroRA often having the fewest or close to the fewest parameters it typically ranks highest

## Weaknesses
The authors only explore smaller ranks for LoRA and AuroRA (a maximum of 16), and while this does demonstrate AuroRA's efficacy in highly parameter-efficient scenarios it doesn't paint a full picture as to how AuroRA compares at larger ranks; for example it is fairly common for fine-tunes of larger models to use ranks as high as 64 or even 256 for more complex tasks, and the paper lacks the experimental evidence for AuroRA's performance in such use-cases.

The author's also seem to lack information in the main body of the paper for important hyperparameters such as alpha which determines the scaling used for LoRA, MoSLoRA and (by confirming in the source code) AuroRA. The alpha choices are listed in the appendix for the primary AuroRA training experiments, but the alpha choices for AuroRA and LoRA variants are unclear for the ablations which show AuoRA to be superior at different parameter counts or ranks. Additionally, there are no ablations over the importance of alpha for AuroRA which is worrying considering sub-optimal choices of alpha for similar PEFT methods can greatly impact convergence.

Finally, I am highly skeptical of the proposed "weight merging" method during inference, where AuroRA switches from $B\sigma(Ax)$ to $B\sigma(A)x$; the purpose of weight merging is to fold PEFT weights into the original model weights to replicate identical behaviour while removing and additional parameters, however due to AuroRA's non-linearity it is not numerical possible to do this. While the results in the Appendix do show empirically that this procedure is still capable of retaining similar performance to the "dynamic" AuroRA implementation it also lacks any real analysis on how this is possible and if it generalises to any arbitrary AuroRA weights. The lack of inclusion of this rather important detail in the main body of the paper is also worrying and also the questions as to if the results in the main body of the paper are from "dynamic" weights or "static" weights which further harms the clarity of the author's claims.

However if we ignore the dubious weight merging and consider the minimal computational impact of dynamic weights even at inference, AuroRA's performance and efficiency is still commendable!

---

> ### Author Rebuttal · Authors · 2025-07-29
>
> We sincerely appreciate your insightful comments and valuable suggestions on our work! Below are our point-by-point responses to the issues you have raised.
>
> ---
>
> > W1: The authors only explore smaller ranks for LoRA and AuroRA (a maximum of 16), and while this does demonstrate AuroRA's efficacy in highly parameter-efficient scenarios it doesn't paint a full picture as to how AuroRA compares at larger ranks; for example it is fairly common for fine-tunes of larger models to use ranks as high as 64 or even 256 for more complex tasks, and the paper lacks the experimental evidence for AuroRA's performance in such use-cases.
>
> Thank you for your very insightful question! To address your concerns, we have conducted additional experiments on **more challenging mathematical tasks**. Using LLaMA2-7B as the base model, we compare the performance of LoRA and AuroRA, **setting the rank for both methods to 64** and the hyperparameter alpha to 128. As shown in Table H, we observe that AuroRA continues to exhibit superior performance even with a higher rank and on more complex tasks. Due to our limited computational resources, we are unable to perform tests on larger-scale models and leave this as an avenue for future work.
>
> *Table H. Performance comparison on the GSM8K and MATH datasets using LLaMA2-7B with a rank of 64.*
>
> | Method           |  GSM8K   |   MATH   |   Avg.   |
> | :--------------- | :------: | :------: | :------: |
> | Full Fine-Tuning |   66.5   |   19.8   |   43.2   |
> | LoRA             |   60.6   |   16.9   |   38.7   |
> | AuroRA           | **65.2** | **18.6** | **41.9** |
>
> ---
>
> > Q1: Did the authors perform alpha tuning to determine the best LoRA, MoSLoRA and AuroRA hyperparameters for the comparison ablations, and if so what values were chosen?
>
> Thank you for your feedback! We would like to clarify that, in accordance with prior works [1,2,3], the hyperparameter alpha is typically set to be equal to the rank or twice the rank. In our experiments, the alpha values for both LoRA and MoSLoRA strictly follow the settings from their respective original papers, which is twice the rank. For example, in the sensitivity analysis on rank, we set the alpha values for LoRA, MoSLoRA, and AuroRA to {4, 8, 16, 32} corresponding to ranks of {2, 4, 8, 16}.
>
> ---
>
> > Q2: Were static or dynamic weights utilised when collecting the inference results in the main body of the paper?
>
> Thank you for your detailed comment! We wish to responsibly point out that all experimental results in the main body of the paper are collected using a "dynamic training, static merging" strategy. As observed in Table 7 (Appendix B.1), when a "fully dynamic" strategy is employed, the AuroRA-D variant actually achieves better performance compared to AuroRA, albeit at the cost of additional runtime. Furthermore, we provide a complete and detailed algorithm for AuroRA in Appendix B.2 to more clearly illustrate the processes during the training and inference stages. Finally, respectfully following your suggestion, we will be sure to emphasize this point more explicitly in the main body of our paper in a future revision.
>
> ---
>
> > Q3: Can the authors justify how it is mathematically possible to linearize AuroRA considering its data-dependent activations and what the implications are considering it re-introduces a linear low-rank bottleneck.
>
> We would first like to clarify the rationale for our "dynamic training, static inference" strategy is to maximize model expressiveness during training while allowing for weight merging to eliminate any additional computational overhead at inference time. **To address your concerns, we provide the following mathematical analysis of this strategy's reasonableness.**
>
> The dynamic and static forms are defined as $h_D = \mathbf{B}\sigma(\mathbf{A}x)$ and $h_S = (\mathbf{B}\sigma(\mathbf{A}))x$, respectively. We aim to demonstrate that once the model converges through dynamic training, the static form used during inference closely approximates the dynamic form, such that $h_D \approx h_S$. Let's consider the Taylor series expansion of our Adaptive Nonlinear Layer (ANL), $\sigma(\cdot)$, around the origin. Let $z = \mathbf{A}x$, we have $\sigma(z) \approx \sigma(0) + \sigma'(0)z + \mathcal{O}(\|z\|^2)$, where $\sigma(0)$ is the function's value and $\sigma'(0)$ is its Jacobian matrix at the origin. As σ is applied element-wise, its Jacobian is a diagonal matrix. For the dynamic form, substituting the expansion gives: $\mathbf{B}\sigma(\mathbf{A}x) \approx \mathbf{B}(\sigma(0) + \sigma'(0)\mathbf{A}x) = \mathbf{B}\sigma(0) + \mathbf{B}\sigma'(0)\mathbf{A}x$. In this case, the $\mathbf{B}\sigma(0)$ term functions as a constant bias, which is independent of the input $x$. For the static form, we analyze the application of $\sigma$ to matrix $\mathbf{A}$. Each element $a_{ij}$ is transformed as $\sigma(a_{ij}) \approx \sigma(0) + \sigma'(0)a_{ij}$. For the entire matrix, this yields $\sigma(\mathbf{A}) \approx \sigma(0) \cdot J + \sigma'(0)\mathbf{A}$, where $J$ is a matrix of ones. The static form then becomes: $(\mathbf{B}\sigma(\mathbf{A}))x \approx \mathbf{B}(\sigma(0)J + \sigma'(0)\mathbf{A})x = \mathbf{B}\sigma(0)Jx + \mathbf{B}\sigma'(0)\mathbf{A}x$. Upon comparison, we observe that the first-order term, $\mathbf{B}\sigma'(0)\mathbf{A}x$, is identical in both forms. For the zeroth-order term, the fixed non-linearity we adopt has $\tanh(0) = 0$, and a negligible bias $\mathcal{L}(0) = \mathbf{w}_s \cdot \mathbf{s}(0)$ exists only in the learnable non-linearity. Overall, we can consider $\sigma(0) \approx 0$. Therefore, the distinction between the dynamic and static forms mainly resides in the higher-order terms. The dynamic form preserves the complete, input-specific non-linear information for each $\mathbf{A}x$, whereas the static form "bakes in" or averages this high-order information into the merged weights. Since all learnable components have been optimized during training, this difference is minor, a point that is also empirically supported by our results in Table 7 of Appendix B.1.
>
> Furthermore, we present an intuitive example in Appendix C, which shows that AuroRA can fit more complex matrix structures thanks to its element-wise non-linear application. This indicates that by transforming the purely linear mapping of the original LoRA into an MLP-like structure, AuroRA can capture more information during training, thereby breaking the low-rank bottleneck.
>
> ---
>
> **Reference:**
>
> [1] LoRA: Low-Rank Adaptation of Large Language Models, ICLR'22
>
> [2] Mixture-of-Subspaces in Low-Rank Adaptation, EMNLP'24
>
> [3] PiSSA: Principal Singular Values and Singular Vectors Adaptation of Large Language Models, NeurIPS'24

---

> > ### Comment · Reviewer_4VGD · 2025-07-31
> >
> > Thank you for addressing weakness 1, it is reassuring to know that AuroRA continues to scale making it a competitive choice over LoRA for a wider variety of ranks.
> >
> > In terms of Q1 you make completely valid points as $\alpha=r$ or $\alpha=2r$ are indeed common choices and I feel confident that such a decision shouldn't affect your paper's chances of acceptance. Despite that I still think it would be worth exploring alpha tuning in future works or a follower up paper; in my experience working with LoRAs it is often necessary to either fix the learning rate but tune alpha for each rank or fix the alpha but tune the learning rate for each rank to obtain the optimal performance for a given task. It would be interesting to see if the gap narrows or widens between LoRA and AuroRA if both are operating at their optimal hyperparameter choices.
> >
> > Thank you for addressing Q2 with your plans to emphasize the use of static AuroRA for evaluations.
> >
> > My biggest concern with the paper was expressed in Q3 as my understanding of mathematics lead me to believe it was not possible to linearise AuroRA due to the non-linearities. However your response makes it much more clear as to how it ***is*** in fact possible to linearise AuroRA to a static approximation which no longer relies on input dependency. I suppose it is a related (but different) phenomena to MoSLoRA, where technically speaking the inclusion of the subspace mixer from a purely analytical standpoint yields no improvement over normal LoRA, but despite that it is often able to outperform LoRA by altering the training dynamics and ultimately converges to more optimal weights; in that regard while AuroRA does introduce non-linearity which is a direct improvement over LoRA, when converted to a static form it does so with more optimal weights than what LoRA converges to which allows it to perform better even without the input dependency.
> >
> > With these considerations I am confident that my rating for AuroRA should be increased.

---

> > > ### Author Response · Authors · 2025-08-01
> > >
> > > Dear Reviewer `4VGD`,
> > >
> > > Thank you very much for your invaluable support of our work! We truly appreciate your insightful comments and your engagement in our clarification. Your recognition of the novelty and effectiveness of AuroRA is also greatly appreciated. Thank you again for your time and dedicated effort in reviewing our paper.
> > >
> > > Best Regards,
> > >
> > > Authors

---

### Official Review · Reviewer_w5vP · 2025-06-29

**Clarity:** 3
**Significance:** 3
**Originality:** 2
**Rating:** 4
**Confidence:** 4

**Summary:**

The paper is about parameter efficient fine tuning in deep learning.  The authors begin with LoRA, which uses an additive linear bottleneck to alter the function of a network, and add non-linearities to the bottleneck to enhance its capability.  In this case the non-linearities comprise an MLP-like layer and a b-spline function.
Results seem pretty good.

**Questions:**

Were the authors able to reproduce the results reported from other literature?

Despite the experimental (ablation) evidence, I'm left wondering from a theoretical point of view why one needs a fixed non-linearity when there is already an adaptive one.  The non-linearities themselves are supposed to be universal approximators; that is why they work in regression at all.

**Ethical Concerns:**

["NO or VERY MINOR ethics concerns only"]

**Final Justification:**

The authors responded to the comments quite well; it would be a better paper with the structural changes.  However, I haven't changed the score as it represents a "gut feeling" in the context of there being lots of PEFT techniques and this being a selective venue.

**Limitations:**

Yes.

**Paper Formatting Concerns:**

Line 194-198, the complexity of matrix multiplication of A (d_out r) and B (r d_in) should be d_out r d_in? I could be wrong.  Applies to all results here.

**Quality:**

2

**Strengths And Weaknesses:**

Strengths:
Novelty: the authors introduce a non-linear LoRA bottleneck with adaptive nonlinear layer, which comprises fixed (with tanh and self-projection) and learnable (B-spline basis functions with learnable weights) nonlinearity.
Effectiveness: the technique matches or surpasses full fine-tuning performance with only 6.18%, being 25% of LoRA’s parameters; impressive performance gains on common sense reasoning tasks with Llama 3 with less than 7% compared to LoRA.
Experimentation: the paper demonstrates comprehensive evaluation on NLP and CV tasks; effectiveness of proposed components verified in ablation studies.
There is a thorough proof of the proposition.

Weaknesses:
Experimentation: for method comparison in the NLP experiments, the authors took results from other literature, which could have been produced under different settings and are hence suboptimal. This weakens the authors’ claim of performance gain.

References:  The citation is a little lazy; first in the way multiple citations are grouped [1,2,3,4,5,...] and secondly in the way some of the literature is after the main content as a kind of afterthought.  This leads to a lot of literature being cited but not discussed well.  That related literature is presented after the main experiments is somewhat unscientific.  It suggests that the experiments were done before doing a thorough literature search, further implying another issue: that the content is perhaps not all that novel.

---

> ### Author Rebuttal · Authors · 2025-07-29
>
> Thank you for your detailed and constructive feedback on our paper. We provide our point-by-point responses to your comments below.
>
> ---
>
> > Q1: Were the authors able to reproduce the results reported from other literature?
>
> To address your concerns, we reproduce the results for all baselines on the Commonsense Reasoning benchmarks using LLaMA3-8B as the base model. The experiments are conducted on an NVIDIA L20 (48GB) GPU, strictly following the hyperparameter configurations from [1]. The results, detailed in Table G, show that the overall outcomes are consistent with those reported in our paper. Furthermore, it is evident that AuroRA achieves the strongest performance with the minimum number of trainable parameters.
>
> *Table G. Reproduction results for LLaMA3-8B on the Commonsense Reasoning benchmarks.*
>
> | Method | Params.  |  BoolQ   |   PIQA   | SIQA     | HellaSwag | WinoGrande |  ARC-e   |  ARC-c   |   OBQA   |   Avg.   |
> | :----- | :------: | :------: | :------: | -------- | :-------: | :--------: | :------: | :------: | :------: | :------: |
> | LoRA   |  56.6M   |   70.8   |   85.1   | **79.7** |   92.0    |    84.3    |   84.2   |   71.5   |   79.0   |   80.8   |
> | PiSSA  |  83.8M   |   67.2   |   81.0   | 77.2     |   83.5    |    78.7    |   77.7   |   63.4   |   74.9   |   75.5   |
> | MiLoRA |  56.6M   |   68.8   |   86.7   | 77.2     |   92.9    |  **85.6**  |   86.8   |   75.5   |   81.8   |   81.9   |
> | AuroRA | **3.5M** | **72.5** | **87.4** | 79.0     | **94.2**  |    83.0    | **89.3** | **78.8** | **84.8** | **83.6** |
>
> ---
>
> > Q2: Despite the experimental (ablation) evidence, I'm left wondering from a theoretical point of view why one needs a fixed non-linearity when there is already an adaptive one. The non-linearities themselves are supposed to be universal approximators; that is why they work in regression at all.
>
> We appreciate this highly insightful question! In line with our proposed "Coarse Fitting" and "Fine Fitting", the fixed non-linearity component serves to stabilize the optimization in its early phases. This establishes a smoother optimization environment that is easier to explore, thereby avoiding the issue of converging to a suboptimal local minimum, which can occur when using a purely learnable non-linearity in very low-rank settings (such as r=2). For a detailed analysis of the approximation error, please refer to Appendix D of our paper.
>
> ---
>
> > References: The citation is a little lazy; first in the way multiple citations are grouped [1,2,3,4,5,...] and secondly in the way some of the literature is after the main content as a kind of afterthought. This leads to a lot of literature being cited but not discussed well. That related literature is presented after the main experiments is somewhat unscientific. It suggests that the experiments were done before doing a thorough literature search, further implying another issue: that the content is perhaps not all that novel.
>
> Thank you for your valuable suggestion! We would like to clarify that our experiments were not conducted before our literature survey. At the same time, we will revise our manuscript following your suggestions. Furthermore, we acknowledge that due to page limitations, we were unable to discuss some related works, and we commit to expanding discussion in a future version of the paper.
>
> ---
>
> > **Paper Formatting Concerns:** Line 194-198, the complexity of matrix multiplication of A (d_out r) and B (r d_in) should be d_out r d_in? I could be wrong. Applies to all results here.
>
> Thank you for your comment! We would like to clarify that there is no error in lines 194-198. You are correct that the complexity of an explicit matrix multiplication of $\mathbf{B}\mathbf{A}\in\mathbb{R}^{d_{\mathrm{out}}\times d_{\mathrm{in}}}$ is indeed $\mathcal{O}(d_{\mathrm{out}}\,r\,d_{\mathrm{in}})$. However, this explicit computation is not performed in LoRA. Instead, the operation is carried out in two sequential steps:
>
> 1. $M = X \mathbf{A}^\top$ is computed, with a complexity of $\mathcal{O}(b\,d_{\mathrm{in}}\,r)$.
> 2. $\Delta \mathcal{W} = M \mathbf{B}^\top$ is computed, with a complexity of $\mathcal{O}(b\,r\,d_{\mathrm{out}})$.
>
> ---
>
> **Reference:**
>
> [1] MiLoRA: Harnessing Minor Singular Components for Parameter-Efficient LLM Finetuning, NAACL'25

---

> > ### Comment · Reviewer_w5vP · 2025-08-04
> >
> > Thank you for addressing the review; the answers are reassuring.
> > Overall, especially in light of the other reviews which are often more detailed than my own, I'm going to stick with the current rating.  I mean it to be generally positive but conveying the reservation that NeurIPS is a selective venue.

---

> > > ### Author Response · Authors · 2025-08-04
> > >
> > > Dear Reviewer `w5vP`,
> > >
> > > We sincerely thank you for your endorsement of our work! We deeply appreciate your valuable feedback and your engagement in our clarification. Your recognition of AuroRA's novelty, effectiveness, and comprehensive experiments is especially encouraging. Thank you again for your dedicated time and effort.
> > >
> > > Best Regards,
> > >
> > > Authors

---

### Official Review · Reviewer_FTUd · 2025-07-01

**Clarity:** 3
**Significance:** 3
**Originality:** 3
**Rating:** 5
**Confidence:** 3

**Summary:**

This paper introduces AuroRA, a novel variant of LoRA (Low-Rank Adaptation) designed to overcome its representational limitations by introducing an Adaptive Nonlinear Layer (ANL) between the low-rank matrices. By incorporating both fixed and learnable nonlinearities, AuroRA forms an MLP-like structure with enhanced expressiveness, better parameter efficiency, and theoretical guarantees for lower approximation error and bounded gradients. The method is evaluated across 22 datasets and 6 pretrained models, spanning both NLP and CV domains, and consistently outperforms existing PEFT baselines including LoRA and its variants.

**Questions:**

1. PEFT methods are generally more useful in fine-tuning tasks, such as instruction following, math, or coding. How is AuroRA's performance in those tasks?

2. In Figure 1, it is shown that AuroRA even outperforms full-parameter fine-tuning. Is there an intuitive explanation for this phenomenon?

**Ethical Concerns:**

["NO or VERY MINOR ethics concerns only"]

**Final Justification:**

The additional baseline comparison results address most of my concerns, so I have increased my quality score and rating.

**Limitations:**

yes

**Paper Formatting Concerns:**

No major formatting issues in this paper are noticed.

**Quality:**

3

**Strengths And Weaknesses:**

### Strengths
1. The paper clearly identifies the low-rank bottleneck of LoRA and proposes a well-motivated nonlinear extension. The shift from purely linear mappings to hybrid nonlinear designs is both novel and practical.

2. The incorporation of fixed (e.g., tanh) and learnable (e.g., spline-based) nonlinearities is well-explained, with theoretical support including proofs for lower approximation error and gradient boundedness.

3. AuroRA achieves good performance in both NLP (e.g., GLUE, Commonsense Reasoning) and CV tasks (e.g., CIFAR10, ViT-based classification), while using significantly fewer parameters than full fine-tuning or LoRA.

4. Extensive ablation studies (e.g., effects of nonlinearity types, activation functions, rank sensitivity) provide convincing evidence of AuroRA’s robustness and component effectiveness.

5. The paper is generally well-written, with clear figures (e.g., PCA visualizations) and structured discussions on design, theory, and evaluation.

### Weakness
1. The paper acknowledges not evaluating on larger models, which would strengthen the claim of scalability.

2. While the paper argues that the additional complexity from ANL is negligible, more concrete runtime/memory comparisons would improve practical understanding.

3. While claiming state-of-the-art, empirical comparisons with recent nonlinear variants (e.g., LoRAN, NEAT) and other PEFT methods are missing [1][2][3][4][5][6][7]. Please remove this state-of-the-art claim if these baselines are not compared.

4. Some missing related works in the PEFT literature: HFT [3], LISA [4], BAdam [5], Owlore [6], Galore [7].

### Reference

[1] Yinqiao Li, Linqi Song, and Hanxu Hou. LoRAN: Improved low-rank adaptation by a non-linear transformation. In Yaser Al-Onaizan, Mohit Bansal, and Yun-Nung Chen, editors, Findings of the Association for Computational Linguistics: EMNLP 2024, pages 3134–3143, Miami, Florida, USA, November 2024. Association for Computational Linguistics.

[2] Yibo Zhong, Haoxiang Jiang, Lincan Li, Ryumei Nakada, Tianci Liu, Linjun Zhang, Huaxiu Yao, and Haoyu Wang. Neat: Nonlinear parameter-efficient adaptation of pre-trained models, 2025.

[3]: Hui, Tingfeng, et al. "Hft: Half fine-tuning for large language models." arXiv preprint arXiv:2404.18466 (2024).

[4]: Pan, Rui, et al. "LISA: layerwise importance sampling for memory-efficient large language model fine-tuning." NeurIPS 2024.

[5]: Luo, Qijun, Hengxu Yu, and Xiao Li. "BAdam: A memory efficient full parameter optimization method for large language models." NeurIPS 2024.

[6]: Li, Pengxiang, et al. "Owlore: Outlier-weighed layerwise sampled low-rank projection for memory-efficient llm fine-tuning." arXiv preprint arXiv:2405.18380 (2024).

[7]: Zhao, Jiawei, et al. "Galore: Memory-efficient llm training by gradient low-rank projection." arXiv preprint arXiv:2403.03507 (2024).

---

> ### Author Rebuttal · Authors · 2025-07-29
>
> We would like to express our sincere appreciation for your time and effort in reviewing our manuscript! Here, we address your comments on a point-by-point basis.
>
> ---
>
> > W1: The paper acknowledges not evaluating on larger models, which would strengthen the claim of scalability.
>
> Thank you for your valuable comment! Due to computational constraints, we are currently unable to test on larger-scale models. We leave this as a promising direction for future work.
>
> ---
>
> > W2: While the paper argues that the additional complexity from ANL is negligible, more concrete runtime/memory comparisons would improve practical understanding.
>
> We appreciate your detailed comments!
>
> We would like to respectfully point out that we provide a detailed comparison of runtimes in Table 7 (Appendix B.1), which shows that AuroRA improves performance without a significant increase in computational time.
>
> Furthermore, to address your concerns, we compare the GPU memory footprint of LoRA and AuroRA on the Commonsense Reasoning task, using the LLaMA3-8B as the base model on NVIDIA L20 (48GB) GPU. As shown in Table C, we observe that AuroRA introduces only a minimal additional memory overhead.
>
> *Table C. Comparison of GPU memory usage on Commonsense Reasoning tasks with LLaMA3-8B.*
>
> | Method | Mem.     |
> | ------ | :------- |
> | LoRA   | 29590MiB |
> | AuroRA | 30834MiB |
>
> ---
>
> > W3: While claiming state-of-the-art, empirical comparisons with recent nonlinear variants (e.g., LoRAN, NEAT) and other PEFT methods are missing [1,2,3,4,5,6,7]. Please remove this state-of-the-art claim if these baselines are not compared.
>
> Thank you for this valuable suggestion!
>
> To demonstrate the performance comparison with recent non-linear variants, we compare AuroRA against NEAT [2] and SineLoRA [8] on the GLUE and Commonsense Reasoning benchmarks, using RoBERTa-Base and LLaMA3-8B as the respective base models. As shown in Tables D and E, the results indicate that AuroRA achieves optimal performance with the fewest parameters.
>
> *Table D. Performance and parameter comparison of RoBERTa-Base on the GLUE benchmark.*
>
> | Method |  Params.   |  SST-2   |   MRPC   |   CoLA   |   QNLI   |   RTE    |  STS-B   |   Avg.   |
> | :----- | :--------: | :------: | :------: | :------: | :------: | :------: | :------: | :------: |
> | LoRA   |    0.3M    |   95.1   |   89.7   |   63.4   |   93.3   |   78.4   | **91.5** |   85.2   |
> | NEAT   |    0.3M    | **95.2** |   90.0   |   64.8   |   92.3   |   82.7   |   90.7   |   86.6   |
> | AuroRA | **0.075M** | **95.2** | **91.9** | **65.1** | **93.4** | **85.2** | **91.5** | **87.1** |
>
> *Table E. Performance and parameter comparison of LLaMA3-8B on Commonsense Reasoning benchmarks.*
>
> | Method   | Params.  |  BoolQ   |   PIQA   |   SIQA   | HellaSwag | WinoGrande |  ARC-e   |  ARC-c   |   OBQA   |   Avg.   |
> | :------- | :------: | :------: | :------: | :------: | :-------: | :--------: | :------: | :------: | :------: | :------: |
> | LoRA     |  56.6M   |   70.8   |   85.2   | **79.9** |   91.7    |    84.3    |   84.2   |   71.2   |   79.0   |   80.8   |
> | SineLoRA |  56.6M   |   72.4   |   86.5   |   79.8   |   94.0    |  **85.2**  |   87.6   |   78.1   | **85.0** | **83.6** |
> | AuroRA   | **3.5M** | **72.5** | **87.4** |   79.0   | **94.2**  |    83.0    | **89.3** | **78.8** |   84.8   | **83.6** |
>
> At the same time, we wish to responsibly point out that, unlike other recent non-linear variants, AuroRA transforms the purely linear structure of vanilla LoRA into an MLP-like structure. This not only introduces non-linearity but also breaks LoRA's inherent low-rank bottleneck.
>
> **We also commit to revising our claims accordingly, citing and expanding our discussion on the other PEFT methods you mentioned [3,4,5,6,7] in a future version of our paper.**
>
> ---
>
> > W4: Some missing related works in the PEFT literature: HFT [3], LISA [4], BAdam [5], Owlore [6], Galore [7].
>
> We appreciate you bringing this to our attention! We will amend the related work section to include these references [3,4,5,6,7] in the final manuscript.
>
> ---
>
> > Q1: PEFT methods are generally more useful in fine-tuning tasks, such as instruction following, math, or coding. How is AuroRA's performance in those tasks?
>
> Thank you for your very insightful question! Firstly, we would like to point out that in Section 3.3, we apply AuroRA to the Subject-driven Image Generation task, **which is a form of multimodal instruction-following**. As demonstrated in Figures 4 and 6, AuroRA exhibits significantly better adherence to prompts compared to vanilla LoRA.
>
> Furthermore, to address your concerns, we conduct extensive experiments on **both math and coding tasks**. The results, presented in Table F, clearly show that AuroRA possesses robust generalization capabilities and maintains superior performance across these diverse tasks.
>
> *Table F. Performance of LLaMA2-7B on math (GSM8K, MATH) and coding (Human-Eval) tasks.*
>
> | Method           |  GSM8K   |   MATH   | Human-Eval |   Avg.   |
> | :--------------- | :------: | :------: | :--------: | :------: |
> | Full Fine-Tuning |   66.5   |   19.8   |    19.9    |   35.4   |
> | LoRA             |   60.6   |   16.9   |    14.8    |   30.8   |
> | AuroRA           | **64.8** | **18.4** |  **22.6**  | **35.3** |
>
> ---
>
> > Q2: In Figure 1, it is shown that AuroRA even outperforms full-parameter fine-tuning. Is there an intuitive explanation for this phenomenon?
>
> We believe this advantage stems from the fact that AuroRA trains only a very small number of parameters (specifically, 6.18% to 25% of those in LoRA, and 0.04% to 0.06% of those in full fine-tuning). This approach prevents the catastrophic forgetting of general knowledge acquired during the pre-training phase. Furthermore, we consider this to be a demonstration of the robust stability of the AuroRA method, as it avoids overfitting to the dataset. To further illustrate this, we provide an intuitive example in Appendix C that highlights the advantages of AuroRA's non-linearity compared to vanilla LoRA.
>
> ---
>
> **Reference:**
>
> [1] LoRAN: Improved low-rank adaptation by a non-linear transformation, EMNLP'24
>
> [2] Neat: Nonlinear Parameter-efficient Adaptation of Pre-trained Models
>
> [3] HFT: Half Fine-Tuning for Large Language Models, ACL'25
>
> [4] LISA: Layerwise Importance Sampling for Memory-Efficient Large Language Model Fine-Tuning, NeurIPS'24
>
> [5] BAdam: A Memory Efficient Full Parameter Optimization Method for Large Language Models, NeurIPS'24
>
> [6] Outlier-weighed Layerwise Sampling for LLM Fine-tuning, ACL Findings'25
>
> [7] Galore: Memory-efficient llm training by gradient low-rank projection, ICML'24
>
> [8] Sine activated low-rank matrices for parameter efficient learning, ICLR'24

---

> > ### Comment · Area_Chair_G3Vf · 2025-08-05
> >
> > Dear Reviewer FTUd,
> >
> > Can you please read the rebuttal and provide your thoughts?
> >
> > Thanks,
> > AC

---

> > ### Comment · Reviewer_FTUd · 2025-08-05
> >
> > Thanks for the response. The additional results addressed some of my concerns. But since most mentioned baseline results and runtime comparison are not available in the rebuttal, I would keep my current score until further updates from the authors.

---

> > > ### Author Response · Authors · 2025-08-06
> > >
> > > Dear Reviewer `FTUd`,
> > >
> > > Thank you very much for your follow-up and feedback!
> > >
> > > First and foremost, following your valuable suggestions, we are committed to revising the manuscript. **We will correct the inappropriate "state-of-the-art" claim and incorporate the citations for the other PEFT methods you mentioned [3,4,5,6,7] in the future version.**
> > >
> > > Furthermore, to address your concerns, we have conducted additional comparative experiments. We compare AuroRA against LoRA, LISA [4], OWS [6], and GaLore [7] on the MMLU benchmark, using LLaMA2-7B as the base model to compare both performance and time cost. **As the results in Table I demonstrate, AuroRA achieves a superior balance between performance and computational efficiency.**
> > >
> > > Thank you once again for the time and effort you have dedicated to our clarification. We sincerely appreciate it.
> > >
> > > *Table I. Performance and time cost comparison of LLaMA2-7B on the MMLU benchmark.*
> > >
> > > | Method     |       Time. | Humanities |   STEM   | Social.  |  Other   |   Avg.   |
> > > | :--------- | ----------: | :--------: | :------: | :------: | :------: | :------: |
> > > | LoRA       | $1.0\times$ |    46.1    |   40.8   |   56.6   |   56.2   |   49.9   |
> > > | GaLore [7] | $4.5\times$ |    45.4    |   41.7   |   55.8   |   56.0   |   49.7   |
> > > | LISA [4]   | $0.7\times$ |    44.9    |   41.2   |   54.7   |   57.6   |   49.6   |
> > > | OWS [6]    | $0.9\times$ |    49.8    |   42.1   |   58.6   | **59.7** |   52.6   |
> > > | AuroRA     | $1.1\times$ |  **49.9**  | **42.8** | **58.9** |   59.5   | **52.8** |
> > >
> > > Best Regards,
> > >
> > > Authors

---

### Official Review · Reviewer_dGxY · 2025-07-02

**Clarity:** 4
**Significance:** 2
**Originality:** 3
**Rating:** 4
**Confidence:** 4

**Summary:**

This paper proposes AuroRA, a novel PEFT method designed to address the low-rank bottleneck inherent in Low-Rank Adaptation. Unlike traditional LoRA or its linear variants that remain constrained by their linear nature, AuroRA introduces a nonlinear mapping through an Adaptive Nonlinear Layer placed between the two low-rank projection matrices. ANL includes both fixed (e.g., tanh) and learnable (e.g., spline-based) nonlinearities, forming an MLP-like structure. This structure enhances expressive power and compresses parameter count while maintaining training stability.The authors support their method with 1) Theoretical analysis, showing reduced approximation error and bounded gradients. 2) Comprehensive experiments across 22 datasets and 6 pretrained models (NLP and CV).

**Questions:**

1 - In the paper, you motivate AuroRA as a means to close the performance gap between LoRA and full fine-tuning by increasing representational capacity. However, in several experimental results (e.g., Table 1 and Table 3), LoRA already performs on par or better than full fine-tuning. Could you provide empirical evidence or experimental configurations where LoRA significantly underperforms full fine-tuning, and AuroRA meaningfully narrows that gap? This would help validate the motivation more clearly.


2 - Several recent nonlinear LoRA variants are cited (e.g., LoRAN, LoDA, NEAT, SineLoRA), but not included in the empirical comparisons. To better contextualize the performance of AuroRA, can you add a comparison against these methods on at least one representative NLP and CV benchmark? This would clarify whether your proposed nonlinear design offers substantive improvements over existing nonlinear extensions to LoRA.

3 - A key advantage of LoRA is that the learned low-rank matrices can be merged into the base weights, eliminating inference overhead. Since AuroRA includes nonlinearities (e.g., tanh, B-spline), this merging is no longer possible. Can you include an inference-time benchmarking comparing AuroRA to LoRA to demonstrate that the additional runtime cost is minimal or acceptable in practice?

**Ethical Concerns:**

["NO or VERY MINOR ethics concerns only"]

**Final Justification:**

After considering the rebuttal, I am keeping my score unchanged (borderline accept).

**Resolved issues**

* The additional CV results (e.g., StanfordCars with ViT) demonstrate cases where LoRA significantly underperforms full fine-tuning, and AuroRA narrows the gap.
* Comparisons to recent nonlinear variants (NEAT, SineLoRA) on representative NLP/CV benchmarks improve empirical grounding.
* The explanation of the “Dynamic Training, Static Inference” strategy and its zero-overhead inference mode addresses most concerns about runtime cost.

**Remaining concerns**

* In the NLP setting, LoRA often matches or exceeds full fine-tuning, so the core motivation of “closing the gap” remains less motivated outside CV in this paper.
* Added nonlinear LoRA comparisons are limited in scope; broader coverage would strengthen claims of superiority.
* Inference strategy details are mostly confined to the appendix; making them more prominent in the main text would improve clarity.

**Limitations:**

yes

**Paper Formatting Concerns:**

None.

**Quality:**

3

**Strengths And Weaknesses:**

**Strengths:**

1 - The key contribution—injecting nonlinear transformations (ANL) into the LoRA framework—is conceptually sound and well-motivated.

2 - The paper provide broad evaluation across tasks (GLUE, commonsense QA, CV benchmarks, DreamBooth) and modalities (text, vision, generation) with strong empirical results.

3 - The ablations are convincing. Sec. 3.4 shows that both fixed and learnable nonlinearities are essential. Also, rank sensitivity study (Figure 5) shows significantly reduced sensitivity to rank vs. LoRA and MoSLoRA, which is a known practical issue for PEFT models.


**Weaknesses:**

1 - My main concern is that I see a mismatch between motivation and experimental setup. The authors repeatedly claim that AuroRA aims to "close the performance gap between LoRA and full fine-tuning" (e.g., Abstract, Introduction) by improving representational power. However, in the reported experiments (e.g., Table 1, Table 3), LoRA already matches or outperforms full fine-tuning in most settings. As such, there is no empirical demonstration that a performance gap exists or that AuroRA helps to close it.

2 - Comparison set omits nonlinear LoRA variants. Recent methods such as LoRAN[89], LoDA[91], SineLoRA[90], and NEAT[92] are mentioned in related work but not evaluated empirically, leaving open how much extra gain AuroRA offers over prior nonlinear attempts.

3 - One of the main benefits of LoRA is the fact that the low-rank update AB can be merged into the weights of the LLM, W’ = W + AB, so LoRA wouldn’t introduce any inference cost or run-time issue after fine-tuning. But in AuroRA, due to the existence of activations in the middle, weights can’t be merged. A rigorous inference latency benchmarking is required to make sure that inference overhead introduced by AuroRA is negligible.

---

> ### Author Rebuttal · Authors · 2025-07-29
>
> We sincerely thank you for your thoughtful and detailed review of our paper. We provide our point-by-point responses to your questions below.
>
> ---
>
> > Q1: In the paper, you motivate AuroRA as a means to close the performance gap between LoRA and full fine-tuning by increasing representational capacity. However, in several experimental results (e.g., Table 1 and Table 3), LoRA already performs on par or better than full fine-tuning. Could you provide empirical evidence or experimental configurations where LoRA significantly underperforms full fine-tuning, and AuroRA meaningfully narrows that gap? This would help validate the motivation more clearly.
>
> Thank you for your insightful comment! We would like to respectfully point out that in the image classification task on StanfordCars, as shown in Table 3 (Section 3.3), our method demonstrates significant improvements.
>
> 1. When using ViT-Base as the backbone model, the accuracy of **Full Fine-Tuning** is **79.8**, while **LoRA** achieves only **45.4**. In contrast, our proposed **AuroRA** reaches an accuracy of **75.7** while using only **12.7% of LoRA's parameters**, successfully narrowing this performance gap.
> 2. The case for ViT-Large is analogous. We observe a significant performance gap where **LoRA's accuracy (73.3)** is considerably lower than that of **Full Fine-Tuning (88.9)**. Our **AuroRA** model substantially mitigates this gap, reaching **82.5** accuracy while using only **12.5% of LoRA's parameters**.
>
> ---
>
> > Q2: Several recent nonlinear LoRA variants are cited (e.g., LoRAN, LoDA, NEAT, SineLoRA), but not included in the empirical comparisons. To better contextualize the performance of AuroRA, can you add a comparison against these methods on at least one representative NLP and CV benchmark? This would clarify whether your proposed nonlinear design offers substantive improvements over existing nonlinear extensions to LoRA.
>
> To address your concern, we compare our method **against two recently proposed non-linear baselines: NEAT [1] and SineLoRA [2]**. We conduct these comparisons on the GLUE and Commonsense Reasoning benchmarks, using RoBERTa-Base and LLaMA3-8B as backbone models, respectively. As the results in Table A and Table B below demonstrate, AuroRA achieves the highest performance with the minimum number of parameters:
>
> *Table A. Performance and parameter comparison of RoBERTa-Base on the GLUE benchmark.*
>
> | Method |  Params.   |  SST-2   |   MRPC   |   CoLA   |   QNLI   |   RTE    |  STS-B   |   Avg.   |
> | :----- | :--------: | :------: | :------: | :------: | :------: | :------: | :------: | :------: |
> | LoRA   |    0.3M    |   95.1   |   89.7   |   63.4   |   93.3   |   78.4   | **91.5** |   85.2   |
> | NEAT   |    0.3M    | **95.2** |   90.0   |   64.8   |   92.3   |   82.7   |   90.7   |   86.6   |
> | AuroRA | **0.075M** | **95.2** | **91.9** | **65.1** | **93.4** | **85.2** | **91.5** | **87.1** |
>
> *Table B. Performance and parameter comparison of LLAMA3-8B on Commonsense Reasoning benchmarks.*
>
> | Method   | Params.  |  BoolQ   |   PIQA   |   SIQA   | HellaSwag | WinoGrande |  ARC-e   |  ARC-c   |   OBQA   |   Avg.   |
> | :------- | :------: | :------: | :------: | :------: | :-------: | :--------: | :------: | :------: | :------: | :------: |
> | LoRA     |  56.6M   |   70.8   |   85.2   | **79.9** |   91.7    |    84.3    |   84.2   |   71.2   |   79.0   |   80.8   |
> | SineLoRA |  56.6M   |   72.4   |   86.5   |   79.8   |   94.0    |  **85.2**  |   87.6   |   78.1   | **85.0** | **83.6** |
> | AuroRA   | **3.5M** | **72.5** | **87.4** |   79.0   | **94.2**  |    83.0    | **89.3** | **78.8** |   84.8   | **83.6** |
>
> ---
>
> > Q3: A key advantage of LoRA is that the learned low-rank matrices can be merged into the base weights, eliminating inference overhead. Since AuroRA includes nonlinearities (e.g., tanh, B-spline), this merging is no longer possible. Can you include an inference-time benchmarking comparing AuroRA to LoRA to demonstrate that the additional runtime cost is minimal or acceptable in practice?
>
> Thank you for your detailed comment! We wish to clarify our methodology, as discussed in Appendix B.1, **AuroRA uses a "Dynamic Training, Static Inference" strategy, allowing for weight-merging that results in zero extra overhead during inference.** We empirically support this choice in Table 7 (Appendix B.1) , which shows that our strategy strikes the optimal balance between accuracy and time consumption when compared to fully dynamic (AuroRA-D), fully static (AuroRA-S), and standard LoRA approaches. For this reason, we adopt this strategy in all experiments in the main body of the paper. Furthermore, we provide a thorough analysis of the computational complexity during training in Section 2.4 and present the complete algorithmic workflow of AuroRA in Appendix B.2 for full transparency.
>
> ---
>
> **Reference:**
>
> [1] Neat: Nonlinear Parameter-efficient Adaptation of Pre-trained Models
>
> [2] Sine activated low-rank matrices for parameter efficient learning, ICLR'24

---

> > ### Comment · Reviewer_dGxY · 2025-08-04
> >
> > Thank you for the thoughtful and detailed rebuttal.
> >
> > The clarifications regarding the CV experiments, comparisons to recent nonlinear variants, and explanation of the static inference strategy are appreciated and help strengthen the paper. These additions improve the framing of the work.
> >
> > That said, some of the original concerns remain partially open. In particular, the motivation around narrowing the gap with full fine-tuning is still less compelling in NLP tasks, where LoRA already performs competitively. While the added comparisons are valuable, broader empirical coverage would further solidify the contribution. Additionally, the treatment of inference-time overhead could benefit from being made more prominent in the main paper.
> >
> > Overall, I believe the paper presents a technically solid and well-motivated idea with promising results. However, given the current limitations, I am keeping my score unchanged.

---

> > > ### Author Response · Authors · 2025-08-04
> > >
> > > Dear Reviewer `dGxY`,
> > >
> > > Thank you very much for your invaluable support of our work! We are sincerely grateful for your detailed and insightful comments, as well as your willingness to engage with us for clarification.
> > >
> > > We would like to further clarify that, unlike LoRA, which requires a higher rank to achieve strong performance, AuroRA remains competitive with full fine-tuning even at a very low rank. For instance, in the NLP task shown in Figure 1 (left), LoRA's performance degrades significantly at a rank of 2, whereas AuroRA surpasses the full fine-tuning. This advantage is further substantiated by the performance comparisons across various ranks in Figure 5.
> > >
> > > We will carefully incorporate all your valuable suggestions into the revised manuscript. We are also incredibly appreciative of your recognition of our work as **technically solid**, **well-motivated**, and **thoroughly evaluated**. Thank you once again for your time and effort in reviewing our paper.
> > >
> > > Best Regards,
> > >
> > > Authors

---

### Note · Authors · 2025-08-12

**Dear Area Chair and Reviewers,**

We are sincerely grateful for the insightful and constructive feedback provided by all four reviewers. It is incredibly encouraging to see our paper's positive aspects being appreciated, such as being **well-motivated** (Reviewers `dGxY`, `FTUd`), **novel** (Reviewers `FTUd`, `w5vP`, `4VGD`), **technically solid** (Reviewers `dGxY`, `FTUd`), and **supported by thorough experimentation** (Reviewers `dGxY`, `FTUd`, `w5vP`, `4VGD`). We also extend our heartfelt thanks to the reviewers for their willingness to engage with us in the clarification process.

We wish to thank all reviewers for the time and effort they have invested in our work, and we are also grateful for the Area Chair's invaluable guidance and support.

Best Regards,

Authors

---

### Decision · Program_Chairs · 2025-09-17

**Decision:**

Accept (spotlight)

**Comment:**

The reviewers wrote that the paper makes a strong and well-motivated contribution by extending LoRA with nonlinear transformations, addressing the known low-rank bottleneck in a novel and practical way. The introduction of both fixed and learnable nonlinearities is clearly explained and supported with the experimental results demonstrating impressive performance across NLP and CV tasks while using significantly fewer parameters than full fine-tuning or LoRA. The evaluation is broad, spanning GLUE, commonsense reasoning, vision benchmarks, and generation, and the ablation studies convincingly validate the importance of each component, including reduced rank sensitivity. Overall, the work is clearly written, thoroughly evaluated, and provides a meaningful advancement in parameter-efficient fine-tuning. Most of the weaknesses pointed out by the reviewers are addressed in the rebuttal. Hence, it will be accepted. We would suggest the authors to address the remaining concerns in the camera ready version.